# CORRELATION AND NAVIGATION IN THE VOCABULARY KEY REPRESENTATION SPACE OF LANGUAGE MODELS

**Letian Peng, Chenyang An, Jingbo Shang**
`{lepeng, c5an, jshang}@ucsd.edu`
University of California, San Diego

## ABSTRACT

Language model (LM) decoding is based on the next-token prediction (NTP) probability distribution. For neural LMs (e.g., Transformer-based), NTP distribution is essentially a softmax-regularized dot product between an encoded input context (*query*) and fixed vocabulary representations (*keys*). In this paper, we study the effect of the key distribution on the NTP distribution, with a focus on whether the similarity between keys will trigger spurious correlations in NTP. Through knowledge-probing tasks, we show that in the NTP distribution, the few top-ranked tokens are typically accurate. However, the middle-ranked prediction is highly biased towards the tokens that are distributionally (not necessarily semantically) similar to these top ones. For instance, if "P" is predicted as the top-1 token, "A"-"Z" will all be ranked high in NTP, no matter whether they can lead to correct decoding results. This hurts the sampling diversity and makes the sampling of correct, long-tail results hopeless and noisy. We attempt to alleviate this issue via a novel in-context method that iteratively pushes the query representation away from explored regions. Specifically, we include the explored decoding results in the context and prompt the LM to generate something else, which encourages the LM to produce a query representation that has small dot products with explored keys. Experiments on knowledge-probing tasks show that our method leads to efficient navigation away from explored keys to correct new keys. We further extend our method to open-ended and chain-of-thought (for reasoning) generation. Experiment results show that ICN contributes to better generation diversity and improved self-consistency voting performance. Finally, we discuss potential training issues caused by the fixed key space together with the challenges and possible ways to address them in future research. Code: https://github.com/KomeijiForce/KeyNavi.

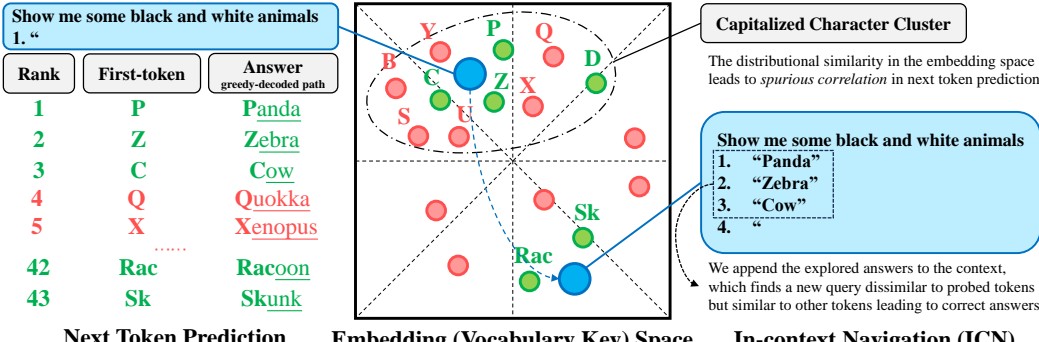

Figure 1: We showcase how next token prediction happens in the key space of the vocabulary embeddings. In the key space, there are clusters, like the capitalized character cluster, containing vocabularies similar to each other. This can introduce spurious correlation in next token prediction because key space similarity is (input) context-agnostic. Tokens with high similarity with top predictions are high-ranked by NTP but might not lead to correct decoding paths. **In-context Navigation (ICN):** We propose a simple in-context method to navigate the query away from the probed keys to efficiently explore the potential decoding paths underestimated by the initial ranking.

# 1 INTRODUCTION

Since the era of statistical language models (LMs) (Brown et al., 1992), LMs have long been decoded by next token prediction (NTP) probability based on statistics from real-world texts. Neural LMs (Bengio et al., 2000; Sutskever et al., 2014; Brown et al., 2020; Dubey et al., 2024) are particularly successful in NTP. Specifically, given an input context, a neural LM first encodes it into a vector (*query*), and then calculates a softmax-regularized dot product between the query and fixed vocabulary representations (*keys*), leading to the predicted NTP distribution.

However, previous works have shown NTP distributions are highly restricted by the arrangement of key embeddings in a context-agnostic manner, such as probability bounds for less frequent words (Demeter et al., 2020) (unpredictable in extreme situations (Grivas et al., 2022)). With the discovery of context-agnostic embedding's limitation in modeling multi-mode word distributions (Chang & McCallum, 2022), we are motivated to explore and mitigate the potential spurious correlation in the embedding space. Different from the major focus on probability bounds in previous works, we emphasize more on distributional precision - is NTP rank a good reference for diverse sampling?

Our investigation begins with the knowledge probing task (Petroni et al., 2019; AlKhamissi et al., 2022; Hao et al., 2022) that extracts factual knowledge from language models. Knowledge-probing requests are typically like "*Show me some black and white animals.*" where the answers are not unique. Based on the request, we decode multiple first-tokens of the potential answer and approximate their *correctness* by whether they can lead to correct answers by greedy decoding or sampling. After probing the top first-tokens predicted by NTP, we visualize the distribution of their key representations. The results show the existence of many incorrect next tokens with high similarity to top (correct) next tokens, which might be introduced by spurious correlation in the key vocabulary space. Thus, we cluster vocabulary embeddings in the key space, which shows many tokens without semantic relation lie in the same cluster. One example is the cluster of capitalized characters, "A"-"Z", which are mutually unrelated when they are subwords in generation. When probing "black and white animals", "P" is predicted as the top-1 token as "P" → "Panda", which leads "Q" also to be high-ranked by its similarity to "P". However, "Q" can hardly lead to a correct answer for "black and white animals", which showcases how spurious correlation in the key space affects the quality of NTP.

To quantify the spurious correlation, we compare the correctness of tokens with and without key space similarity to top-ranked tokens. For 60 probing prompts expanded from CGExpan (Zhang et al., 2020), we group the next tokens ranked among 11-100 by whether they fall in the same clusters as the top-10 tokens (in-top-cluster). Our result shows tokens in different clusters from the top-10 tokens (out-of-top-cluster) have higher accuracy than those in-top-cluster tokens, but are ranked lower by NTP. This indicates that some in-top-cluster tokens are overestimated by spurious correlation in the key space rather than really leading to correct decoding paths.

To alleviate such spurious correlation, we propose a simple yet effective method, in-context navigation (ICN), to efficiently push the query away from explored keys. Specifically, we explicitly append answers starting with probed first-tokens to the context and instruct the LM to generate different answers. Following the instruction, NTP will eliminate the probability of explored tokens, resulting in a low similarity between the new query and the explored keys. Consequently, this simple modification pushes the new query representation away from probed clusters to explore new ones containing the correct first-tokens. In comparison with simple rephrasing the prompt, ICN produces queries dissimilar from explored keys, which reflects a strong pushing-away ability of ICN.

We further benchmark the precision of ICN-based knowledge probing. By iteratively producing new queries away from explored spaces, we discover a significant precision improvement in knowledge probing. We continue to extend this method to explore the potential first-token generation for open-ended generation (Ye et al., 2022c;a) and chain-of-thought generation (Wei et al., 2022) for self-consistency (Wang et al., 2022). The results show higher generation diversity and reasoning accuracy than simply exploring top first-tokens.

Finally, we discuss training risks that might be caused by the fixed key space. By comparing the key space of large LMs before and after the large-scale fine-tuning, we observe the key space is almost unchanged, indicating the key representations have converged at the very early stage. This indicates that fine-tuning the language model is only learning the query encoder to push the query toward a certain key, which unfortunately also pushes the query closer to incorrect keys similar in the

key space. Our quantitative experiment shows when fine-tuning a correct token, the knowledge is generalized to tokens in the same cluster (by increasing their probability) rather than other correct tokens. Thus, we also propose potential refinement methods for future works to further address the spurious correlation in inference and training in this paper. Our contributions are presented as follows,

- We unveil the spurious correlation in the vocabulary key space for NTP, which introduces incorrect tokens into the prediction by their context-agnostic similarity in the key space.
- We propose a simple method, in-context navigation (ICN), to mitigate the spurious correlation using explicit context to search for new queries away from explored keys. ICN is beneficial to knowledge probing, open-ended generation, and chain-of-thought generation.
- We extend the discussion to large-scale model training, revealing that the early-converged key space remains unchanged even during extensive fine-tuning. This also posts a question about the generalization ability of NTP learning - to similar keys or to correct keys?

## 2 RELATED WORK

### 2.1 LANGUAGE MODELING

Language modeling has been an essential problem in a long history of natural language processing, which has attracted more and more attention from both industry and academia since the boom of large language models (Brown et al., 2020; Achiam et al., 2023; Touvron et al., 2023; Team et al., 2024). Similarly to their statistical ancestors Brown et al. (1992), these large neural language models are generally decoded from a probabilistic view, such as random sampling, greedy decoding, beam search, nucleus decoding (Holtzman et al., 2019; Welleck et al., 2024) and contrastive decoding (Li et al., 2023) that exploits predictions from weaker LMs.

However, there is also evidence indicating the limitation of the probabilistic view. The effectiveness of self-consistency (Wang et al., 2022) shows that the higher probability decoding path is not generally more correct. Wang & Zhou also presents that probing the first-token even with a very low probability might lead to a correct answer. Unfortunately, LMs might not be able to model such diverse decoding paths because of the softmax bottleneck (Yang et al., 2018) as they are inclined to monomodal words (Chang & McCallum, 2022). The probabilities of less frequent words are also likely to be bounded when falling inside convex hulls in the distribution (Demeter et al., 2020), causing some words to be unpredictable (Grivas et al., 2022). Based on previous efforts, our work further illustrates and mitigates the potential spurious correlation phenomenon in NTP.

### 2.2 VOCABULARY KEY SPACE OF LANGUAGE MODEL

In language models, the next tokens are represented by fixed vectors to calculate their similarity with encoded context, which is similar to dense retrieval in information retrieval. Dense retrieval (Karpukhin et al., 2020; Xiong et al., 2020) is an essential technique to retrieve relevant documents according to user's request, which encodes requests&documents into query&key representations. When a user request comes, it is first transformed into a query representation whose dot product values with key representations are ranked to determine the most relevant documents. Next token prediction in neural language models can also be discussed from this view that the context is encoded into a query representation, whose dot product values with the vocabulary's key representations determine the most possible next tokens (Cao et al., 2024). There are several trials to improve language modeling by modifying the key space. Adapting key representations by vector transformation is proposed to steer generation (Han et al., 2024). Phrase-augmented language models (Lan et al., 2023; Cao et al., 2024) whose vocabulary is augmented by encoded phrases have been proposed for a more diverse open-ended generation. Our work is inspired by the success of language modeling as retrieval and delves into how vocabulary embeddings are distributed.

### 2.3 DECODING PATH PROBING

There are generally multiple decoding paths that lead to correct responses (Wang et al., 2022). Probing these paths is important for harvesting knowledge, sampling diverse passages, and exploring multiple ways of reasoning from language models. Knowledge-probing (Petroni et al., 2019; AlKhamissi et al., 2022; Hao et al., 2022) prompts language models with factual questions and collects the top

answers as knowledge. Sampling diverse passages is important for data synthesis like training a text classifier (Ye et al., 2022a;b). Voting with diverse chain-of-thought reasoning paths is also promising in improving the reasoning accuracy (Wang et al., 2022; Wang & Zhou, 2024). There are also trials to encode the feature of contexts with the possible answer paths according to questions (Peng et al., 2024; Benara et al., 2024). Our work explores how path probing is affected by the potential correlation in the key space and proposes corresponding methods to address the issue.

## 3 PRELIMINARY AND MOTIVATION

Language models $LM(\cdot)$ are trained to make NTP which produces the probability distribution of the next token to generate. Given an input context $X = [x_1, x_2, \cdots, x_N]$, where $n = |X|$ represents the number of input tokens. The language model $LM(X)$ outputs a probabilistic distribution $P(x_{n+1}|X)$ of the vocabulary $V$ with $|V|$ probabilities of all vocabularies as the next token.

For neural language models, the NTP probability is essentially produced by representation calculation. Inside the language model, the encoder $E(\cdot)$ encodes $X$ into a $D$-dimension representation $R_X$, which is the same as the vocabulary representations $R_V \in \mathcal{R}^{|V| \times D}$. The dot product between $R_X$ and $R_V$ produces a logit $L \in \mathcal{R}^{|V|}$, which is regularized by a softmax function to finally output the probabilistic distribution $P(x_{n+1}|X)$.

The fixed key representations might introduce unexpected *spurious correlation* into NTP because the vocabularies are assigned with context-agnostic similarity. For instance, "P" and "Q" generally show high similarity in the key spaces of different LMs because they are both capitalized characters. However, this should be viewed as a spurious correlation in many contexts (e.g. generating subwords). In the following sections, we will use experiments to visualize and quantify the severity of this issue.

## 4 SPURIOUS KEY CORRELATION

### 4.1 EXPERIMENT SETUP

Knowledge probing (Petroni et al., 2019; AlKhamissi et al., 2022; Hao et al., 2022) is a task that aims to extract as much as possible knowledge from LMs (e.g. "*Show me some computer scientists.*"). We select the knowledge probing task to investigate the spurious correlation issue because there are many first-tokens that can lead to correct decoded answers and the answers are easy to validate. Our probing experiment aims to unveil how the correct next tokens are distributed in the key space and how the distribution affects the NTP result. We illustrate the result on a strong open-source LM, `llama-3-8b-instruct` (Dubey et al., 2024), and include the result on other LMs in Appendix H.

We follow Zhang et al. (2020) for the probing targets, and extend its coverage into 12 categories, such as "Scientist", "Astronomical Object", and "Sports League". To further challenge LLM, we broaden the comprehensiveness of the probing by adding 4 extra sub-categories for each category ($12 + 4 \times 12 = 60$ probing categories in total). For instance, "Computer Scientist" is included as an extra sub-category probing experiment for "Scientist" probing. In the experiment, we refer to the expanded knowledge probing set as

| Scientist | Food | City |
|---|---|---|
| Computer Scientist | Chinese Food | Canadian City |
| Social Scientist | Spicy Food | Asian City |
| Female Scientist | Veggie Food | Coastal City |
| German Scientist | Dessert | Capital City |

Table 1: Examples of categories and sub-categories. Full list can be found in Appendix C.

**ProbeSet**. Some categories and sub-categories involved in our experiments are presented in Table 1. We explore the NTP of the first-token in the answer and use greedily decoded answers (Monte-Carlo sampling results also included in Appendix E) to approximate the decoding space for experiment efficiency like in Wang & Zhou (2024). For validation, we prompt the state-of-the-art LM, `GPT-4o`, whose accuracy is verified to be around $95\%$ by annotated reference in Zhang et al. (2020) and human consistency evaluation shown in Appendix B. The prompts used can be found in Appendix K.

### 4.2 CASE VISUALIZATION

The first stage in our experiment is to visualize the distribution of predicted first-tokens in the key space. Thus, we select three cases, "Scientists", "Astronomical Objectives", "Scientists", to visualize

how correct/incorrect next tokens are distributed. We validate the top-100 predicted next tokens and apply t-distributed Stochastic Neighbor Embedding (t-SNE) (Van der Maaten & Hinton, 2008) to reduce the dimensions of corresponding key representations together with the query representation (encoded context). We select t-SNE because of its ability to maintain the cosine similarity relationship in the high dimension space.

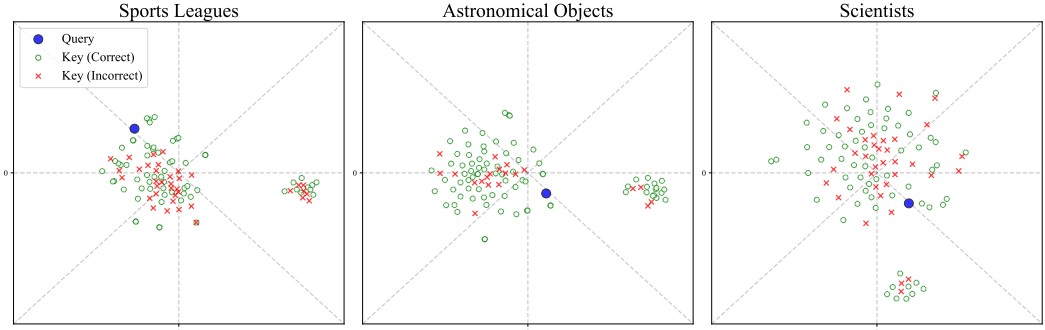

Figure 2: Visualization of the relationship between key representations of first-tokens with their probing correctness in knowledge probing cases.

In Figure 2, we illustrate the visualization of different probing cases. We can observe the query to be encoded near several correct key representations leading to decoding paths to correct answers, which correspond to the top next tokens in NTP. However, there are also incorrect next tokens with high similarity with these top tokens, which are consequently closer to the query representation than other correct tokens. This supports the hypothesis of spurious correlation in NTP, we will further verify the issue existence via metric quantification.

## 4.3 ISSUE QUANTIFICATION

We first propose a metric to describe how NTP is impacted by the spurious correlation in the key space. The ultimate goal of our metric is to depict the difference between the next tokens that are spuriously correlated to top tokens and those aren't. The spurious correlation is independent on the query and only dependent on how keys are similar to one other. Thus, we run a clustering algorithm on the key representations to divide the vocabularies into 1024 clusters. Specifically, we select the K-means (Lloyd, 1982) algorithm because it outputs clusters in the same size, which indicates the same amount

| CID | In-Cluster subwords |
|-----|---------------------|
| 896 | A, B, C, D, E, F, G, H, I, J, K, L, M, N, O, ... |
| 200 | 0, 1, 2, 3, 4, 5, 6, 7, 8, 9, 00, 20, 10, 201, ... |
| 640 | Ġwonder, Ġbeautiful, Ġamazing, Ġexcellent, ... |
| 996 | urn, Ġecho, printf, echo, Ġprintf, ĉecho, ... |
| 100 | elf, eld, elp, els, EL, elt, elves, ael, El, elay, ... |
| 350 | lob, Ġlunch, Ġlens, Ġlip, Ġlobby, Ġlaptop, ... |

Table 2: Examples of the next token clusters in key space clustering of LLaMA-3. CID: The cluster identifier for reference in the paper. "Ġ" and "ĉ" are special tokens, which are decoded into blanks.

of spurious correlation. We showcase some clusters in Table 2 with corresponding in-cluster subwords. The clusters include capitalized characters from "A" to "Z" (CID=896) , and numbers from "0" to "9" (CID=896). There are highly explainable clusters like positive adjectives (CID=640) but there are also clusters without a valid reason for similarity, especially for subwords (CID=100).

Based on the clusters, we design the metric to quantitatively analyze the spurious correlation. Our focus is on the middle-ranked next tokens that are ranked high but not top in NTP, specifically, from all tokens in top-$N$ ($N = 100$) but not in top-$K$ ($K = 10$). The top-$K$ next tokens are viewed as the top tokens that might inject the spurious correlation. The middle-ranked tokens falling in the same cluster as the top tokens are supposed to be affected by the spurious correlation. The middle-ranked tokens are thus divided into two groups: in-top-cluster (InTop) and out-of-top-cluster (OutTop).

$$\text{InTop} = \{v_i | \exists_j (C(v_i) = C(v_j) \land j \le K) \land (K < i \le N)\}$$
$$\text{OutTop} = \{v_i | (v_i \notin \text{InTop}) \land (K < i \le N)\}$$

where $C(\cdot)$ returns the cluster of the $i$-th (ranked by NTP) token $v_i$. We use accuracy and average rank to compare them in performance and distribution. Besides, we illustrate the proportion of the two groups to show the results are not based on very few data points.

| Category | Sports League | | Astronomical | | Scientist | | Landmark | | Country | | City | |
|---|---|---|---|---|---|---|---|---|---|---|---|---|
| Group | InTop | OutTop | InTop | OutTop | InTop | OutTop | InTop | OutTop | InTop | OutTop | InTop | OutTop |
| Accuracy | 50.12 | **60.36** | 49.21 | **71.80** | 50.62 | **64.17** | 70.64 | **76.65** | 61.66 | **68.61** | 65.50 | **69.87** |
| Rank | 54.01 | 56.97 | 54.42 | 56.65 | 52.70 | 58.51 | 54.79 | 56.10 | 54.69 | 55.71 | 52.00 | 59.29 |
| Proportion | 50.89 | 49.11 | 54.00 | 46.00 | 53.11 | 46.89 | 48.67 | 51.33 | 41.78 | 58.22 | 51.78 | 48.22 |

| Category | TV Channel | | Restaurant | | Company | | Creature | | Disease | | Food | |
|---|---|---|---|---|---|---|---|---|---|---|---|---|
| Group | InTop | OutTop | InTop | OutTop | InTop | OutTop | InTop | OutTop | InTop | OutTop | InTop | OutTop |
| Accuracy | 60.33 | **62.97** | 71.15 | **75.97** | 76.41 | **81.00** | 72.87 | **77.66** | 65.55 | **71.44** | 68.00 | **73.66** |
| Rank | 54.59 | 56.96 | 52.03 | 59.39 | 52.04 | 59.82 | 50.54 | 61.69 | 50.50 | 59.64 | 50.07 | 59.87 |
| Proportion | 46.89 | 53.11 | 51.11 | 48.89 | 54.00 | 46.00 | 56.44 | 43.56 | 46.22 | 53.78 | 46.00 | 54.00 |

Table 3: Knowledge probing results on ProbeSet. The shown metrics are calculated by an average over 1 main category + 4 sub-categories. We also provide experiments in probing "words starting with given characters", whose result is independent of LLM evaluator in Appendix D.

We present the knowledge probing result in Table 3 across the 12 categories, which shows that the OutTop first-tokens not only have better accuracy but are ranked lower as well. Thus, the division verifies the existence of spurious correlation, which leads a group of less accurate tokens to be ranked higher. The balanced proportion indicates this not to be an issue stemming from just a few isolated cases, but a general phenomenon that hinders the NTP. Following the observation above, we will discuss potential ways to mitigate the spurious correlation in the following sections.

## 5 IN-CONTEXT NAVIGATION

### 5.1 METHODOLOGY

From previous experiments, the middle-ranked tokens are shown to be the victims of spurious correlation. Thus, our strategy is to decode the LM multiple times, each time only for the relatively accurate top tokens. A simple method is to rephrase the prompt, which perturbs the query in the representation space in search for new top tokens. Our proposed method, in-context navigation (ICN), inherits this idea and steps further to navigate the query representation away from the explored key representations with an explicit instruction in the context. For instance, when we have explored "Alan Turing" for "Computer Scientist", a prompt with ICN can be "A computer scientist other than

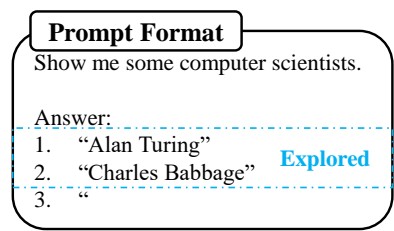

Figure 3: The prompt format used for ICN in knowledge probing experiments.

Alan Turing is", which discourages the LM to generate "Alan Turing" and consequently eliminates the similarity between the query representation and the key representation of "Alan". In our experiments, we use the format shown in Figure 3 to handle a long list of explored keys.

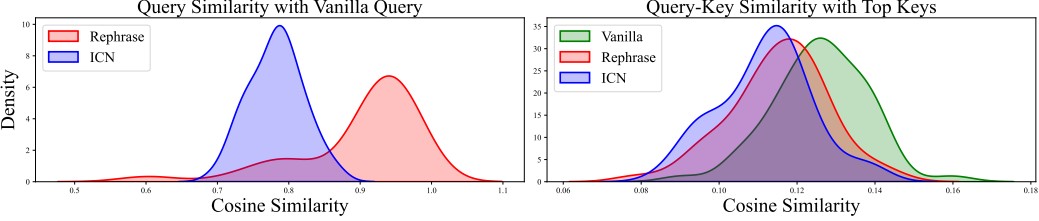

Figure 4: Exploration of the navigation ability of ICN. **Left:** Query similarity with the original query representation. **Right:** Query-key similarity with the top key representations corresponding to the original query. **Rephrasing:** Some rephrasing cases are shown in Appendix K.

We first conduct experiments to certificate the navigation ability of ICN based on ProbeSet. For each instruction, we decode the top-10 first-tokens and append the decoded result to the context, which is encoded to the new query representation. We evaluate two types of similarity: 1) the similarity with the original query representation and 2) the average similarity with the top-10 key representations corresponding to the original query. We include simply rephrasing the probing prompt as a baseline for comparison. The results are plotted in Figure 4 by the distribution curve. The left subfigure illustrates the query similarity, which shows the query is successfully navigated away from the

original one by ICN while simple rephrasing still leads the new query to a position near the original one (with cosine similarity close to 1.0). The right subfigure shows ICN to be better at navigating the query away from explored keys in comparison with the simple rephrasing. Thus, we certificate the ability of ICN to navigate the query to a different location away from probed keys.

The next step is to verify the accuracy of the navigated query, i.e. the correctness of new top first-tokens. Specifically, we compare the probing accuracy between direct decoding and decoding with ICN. To utilize ICN, we introduce the iterative ICN to traverse through key representations. In each iteration, we will decode the path for top first-tokens and append them to the context, which is encoded to a new query for the next iteration. Also, the probed first-tokens are skipped in future iterations as each first-token should be probed only once for comparison.

For iterative ICN, the two key parameters are the number of encoded queries (#Query) and the number of probed top keys (#Key) in each iteration, whose multiplication (#Query $\times$ #Key) should be equal to the number of probed paths. When #Query $= 1$, the procedure equals to direct decoding, which is taken as the baseline result. When #Key $= 1$, the procedure equals to prompting the LM to list the answers.

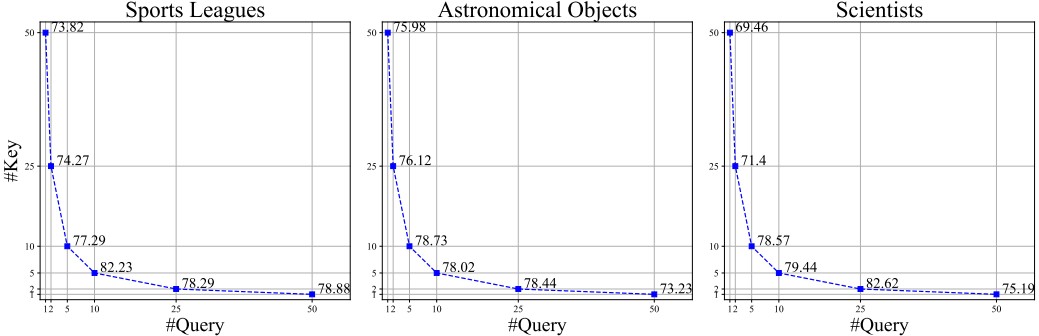

Figure 5: Exploring the impact of ICN frequency on knowledge probing.

We depict probing result of different (#Query, #Key) pair for iterative ICN in Figure 5. We set the number of probing path to be 50 and present the MAP@50 (Everingham et al., 2010) value as the metric, which considers the rank of correct predictions. For comparison fairness, all probed first-tokens by navigated queries are appended back to the initial prompt. Thus, the generation spaces are kept the same for different experiment configurations.

The experiment results show the probing performance with ICN to outperform direct decoding, which indicates ICN to be able to navigate the query close to correct keys. On the other hand, navigating queries too frequently (like listing the answers when #Key$= 1$) also does not lead to the best performance, which indicates the importance of probing multiple top tokens. In general, the best performance is achieved with a balanced value between #Key and #Query, for instance, #Query $= 10$ and #Key $= 5$. In conclusion, navigating queries away from explored keys by ICN is beneficial to mitigate the spurious correlation.

## 5.2 MAIN COMPARISON

Based on the verification of ICN's effect, we further systematically use ProbeSet to benchmark the probing performance of different methods against the spurious correlation in the key vocabulary space. We continue using the ProbeSet for benchmarking. We still probe 50 decoding paths by using MAP@50 and precision as the metric. Based on the conclusion in the probing experiment, we set #Query to 10 and #Key to 5. Similar with previous experiments, the first-tokens probed by ICN or other methods are appended to the initial probing prompt to eliminate the influence of extra context towards a fair comparison.

**Baselines** We include the **vanilla** method, which simply probes decoding paths of top-50 first-tokens. The **rephrasing** method mentioned in the navigation ability evaluation replaces the appending explored paths in ICN by rephrasing the probing instruction to navigate to different first-tokens. We rephrase each instruction for 10 times and probe top-5 (not probed) keys. Another **reranking** baseline penalizes tokens by the number of probed tokens in the same cluster. Specifically, the rank of each token will be added by its rank in its cluster. For instance, when the first 5 tokens are in cluster 1, 1,

1, 2, 2, their in-cluster ranks should be 1, 2, 3, 1, 2. Consequently, the added ranks will be 2, 4, 6, 5, 7, which switch the rank between the initial third and fourth tokens. When the added ranks are equal, the initial rank is considered as the prior one in our implementation. The reranking method explores whether we can eliminate the spurious correlation without searching for multiple queries.

| Category | Sports League | | Astronomical | | Scientist | | Landmark | | Country | | City | |
|---|---|---|---|---|---|---|---|---|---|---|---|---|
| Metric | MAP | PREC | MAP | PREC | MAP | PREC | MAP | PREC | MAP | PREC | MAP | PREC |
| Vanilla | 73.82 | 60.40 | 75.98 | 72.80 | 69.46 | 62.40 | 84.48 | 79.60 | 81.45 | 74.40 | 91.57 | 85.20 |
| Rephrase | 73.78 | 59.60 | 76.69 | **74.00** | 72.15 | 62.80 | 84.31 | 79.60 | 81.94 | 74.80 | 92.05 | 85.60 |
| Rerank | 74.82 | 61.00 | 77.57 | **74.00** | 70.79 | 63.20 | 84.82 | 80.40 | 81.81 | 75.40 | 92.42 | 85.00 |
| ICN | **82.23** | **71.60** | **78.02** | 68.40 | **79.44** | **76.40** | **89.44** | **84.00** | **92.57** | **82.80** | **96.38** | **93.20** |
| Category | TV Channel | | Restaurant | | Company | | Creature | | Disease | | Food | |
| Metric | MAP | PREC | MAP | PREC | MAP | PREC | MAP | PREC | MAP | PREC | MAP | PREC |
| Vanilla | 79.56 | 71.20 | 88.80 | 81.60 | 87.63 | 82.80 | 95.73 | 91.00 | 85.55 | 77.20 | 81.43 | 76.40 |
| Rephrase | 80.87 | 72.00 | 89.35 | 81.60 | 86.83 | 81.60 | 89.59 | 84.80 | 85.48 | 76.80 | 80.90 | 76.80 |
| Rerank | 79.86 | 71.20 | 89.60 | 83.40 | 88.23 | 84.60 | 96.19 | 90.80 | 85.96 | 78.60 | 82.16 | 78.00 |
| ICN | **87.70** | **82.80** | **93.70** | **84.00** | **97.60** | **94.40** | **97.60** | **92.00** | **88.37** | **78.80** | **93.53** | **91.20** |

Table 4: Knowledge probing results with different methods to mitigate spurious correlation.

The experiment result in Table 4 shows that ICN significantly outperforms the vanilla probing strategy, which is consistent with the conclusion in the probing experiment. Rephrasing generally does not show much difference with the vanilla method (sometimes higher and sometimes lower), suggesting it not able to navigate the query to new correct keys. This is consistent with the indication from the similarity experiment, which shows a high similarity between the initial and the rephrased query. Finally, the reranking method achieves small yet consistent improvement across all experiments, which again verifies the existence of spurious correlation. However, only reranking significantly underperforms ICN, which emphasizes the importance of multiple queries. Still, the efficiency advantage from single time encoding maintains the usage of reranking for efficiency.

## 5.3 OPEN-ENDED GENERATION

We further explore the usage of ICN beyond knowledge probing. The first task is open-ended generation, which differs from knowledge probing by generating sentences instead of entities. Thus, the framework of ICN is kept with the instruction changed to generate sentences. The evaluation concentrates on the diversity and usage of the generated sentences. For diversity, we select unique n-gram (UNG) (Buck et al., 2014) as the metric, which counts the proportion of unique n-grams, averaged over $n = 1 \sim 4$. For usage, we evaluate the classifier trained on the generated texts on test sets annotated by humans. This scenario has been proposed as ZeroGen (Ye et al., 2022b), so we name the metric as ZeroGen accuracy (ZGN). As more diverse datasets train better classifiers (Peng & Shang, 2024), the result simultaneously reflects the semantic diversity of text generation.

We select 3 datasets for evaluation, SST-2 (positive, negative) (Socher et al., 2013), AG-News (World, Sports, Business, SciTech) (Zhang et al., 2015), Emotion (Sadness, Joy, Anger, Fear, Love, Surprise) (Saravia et al., 2018). Baselines include repetitively sampling sentences from the prompt, which is generally applied in existing text generation scenarios. Another baseline probed top first-tokens, which corresponds to the vanilla method in the knowledge probing experiments. All methods generate 100 sentences for diversity evaluation and classifier training (selected as RoBERTa-Large, hyperparameters listed in Appendix K). For ICN, we set #Query to 10 and #Key to 10.

The results are presented in Table 5, we can observe the ICN achieving consistent improvement on text generation with better UNG diversity and ZGN accuracy. In comparison with repetitive sampling, probing different first-tokens shows better performance, which indicates the influence of the first-tokens in generation even for sequences

| Method | SST-2 | | AG-News | | Emotion | |
|---|---|---|---|---|---|---|
| | UNG | ZGN | UNG | ZGN | UNG | ZGN |
| Sample | 58.95 | 83.20 | 48.17 | 38.37 | 41.39 | 38.37 |
| Top | 61.29 | 84.90 | 50.97 | 40.63 | 50.85 | 40.63 |
| ICN | **64.74** | **86.33** | **53.37** | **42.19** | **53.34** | **42.19** |

Table 5: Open-ended generation results. "Top" explores decoding paths with top predicted first tokens.

(sentences) longer than entities. The advantage of ICN over simple top probing can be similarly explained as the diverse yet correct first-tokens explored by ICN leading to more diverse sequence generation.

### 5.4 CHAIN-OF-THOUGHT GENERATION

Chain-of-Thought (CoT) (Wei et al., 2022) refers to the reasoning chains in complex tasks such as math problem solving. Similar to open-ended generation, diversity influences the success rate of large model reasoning (Naik et al., 2023) when answers from different CoTs are merged for self-consistency (Wang et al., 2022). When multiple CoTs are generally sampled by multiple times from large languages, Wang & Zhou propose a better way to probe the reasoning path starting from different first-tokens. Their strategy probes the top tokens, similar to the baseline setup in our knowledge probing experiments. Consequently, we apply ICN to this CoT generation framework by appending probed CoTs in the context to probe new first-tokens. (Remind that explored CoTs will be removed when generating the new CoT, which prevents copying the answer in explored CoTs.)

For benchmarking, we use 3 math problem solving datasets from the initial CoT experiments (Wei et al., 2022), which are GSM8K (Cobbe et al., 2021), SVAMP (Patel et al., 2021), and AQuA (Deep-Mind, 2017). GSM8K and SVAMP directly ask for numeric answers while AQuA contains multiple-choice math questions, which select answers from 5 candidates. For self-consistency, we set the number of CoTs to 4. The hyperparameters for ICN are set to #Query = 4 and #Key = 1. For self-consistency, the most voted answer is selected as the final answer.

In Table 6, we illustrate the reasoning performance of self-consistency with different strategies. Consistent with results on previous tasks, our ICN contributes more to self-consistency than sampling and probing only top tokens by proposing diverse and accurate CoTs, which is verified by all 3 datasets. While the first generated tokens might be considered to have a limited impact on the whole CoT quality, our result (together with Wang & Zhou (2024)) suggests the benefit in probing them. On

| Method | GSM8K | SVAMP | AQuA |
|--------|-------|-------|------|
| w/o SC | 74.15 | 79.00 | 53.54 |
| Sample | 76.50 | 83.10 | 55.51 |
| Top | 77.56 | 84.20 | 57.09 |
| ICN | **78.09** | **85.10** | **58.66** |

Table 6: Reasoning benchmark results. SC: Self-Consistency

the other hand, the benefit from ICN to CoT generation is not as significant as knowledge probing, which indicates lengthy generation might weaken the benefit from ICN.

## 6 LM TRAINING RISKS FROM FIXED KEY SPACE

Our previous contents mainly concentrate on the impact of key space during the inference time. In this section, we will dive deeper into the potential influence of the query-key matching procedure during training neural language models.

We first illustrate an important property of the key space, its convergence after the large scale pre-training. Specifically, we compare the key spaces between `llama-3-8b` and `llama-3-8b-instruct`. `llama-3-8b-instruct` is based on `llama-3-8b` with further supervised fine-tuning (SFT) and reinforcement learning with human feedback (RLHF). We evaluate the similarity in three scenarios, 1) **Token Similarity**, calculating the cosine similarity between key representations of the same vocabulary (before or after SFT&RLHF), which is applied to all vocabularies, 2) **Pair Similarity Difference**, calculating the difference of cosine similarity between the same pair of vocabularies (before or after SFT&RLHF), which is applied to 100,000 randomly sampled pairs. 3) **Similarity Rank Difference**, calculating the similarity between a token with all vocabulary tokens, then calculating the spearman correlation between the similarity distribution of the same vocabularies (before or after SFT&RLHF), which is applied to 10,000 randomly sampled vocabularies.

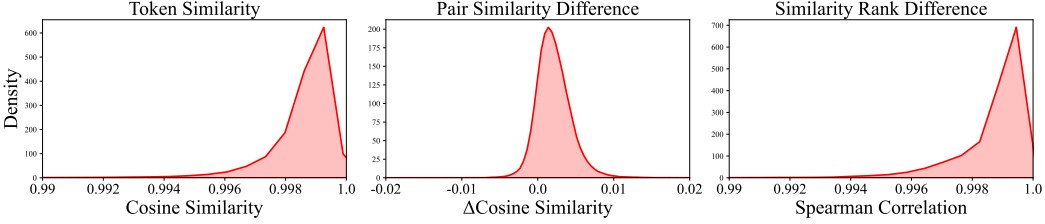

Figure 6: Similarity between `llama-3-8b` and `llama-3-8b-instruct` in key spaces.

As the 3 evaluation results presented in Figure 6, we observe the key spaces before and after SFT&RLHF have very high **Token Similarity** and **Similarity Rank Difference**, together with almost zero **Pair Similarity Difference**. This indicates the key space hardly changed after SFT&RLHF, even though these stages also include numerous training data. A highly possible explanation is the shallow network (only an embedding layer) to encode vocabularies can only capture some spurious correlation between them as shown in Table 2, ignoring the complex interaction between queries and keys in language modeling. Given the early-converged key space and the high performance difference between the models before and after SFT&RLHF, we conclude the fine-tuning stages are mainly learned to encode the context to queries but hardly adjust the key space.

Based on the conclusion above, we would like to further point out the potential vulnerability in fine-tuning large models. As only queries are effectively adjusted, the ability of language models to store multiple knowledge is questionable, especially when two correct NTP answers are in different vocabulary clusters. We quantify this question as "When a correct next token is used for fine-tuning, it is generalizing to (increasing the probability of) other correct next tokens or generalizing to other next tokens in the same cluster?" To answer this question, we go back to the knowledge-probing task and fine-tune (learning rate is set to $10^{-6}$) the large model on the correct top-10 next tokens. For each optimization step, we calculate the probability sum difference between groups of tokens. For comparison, the first group of tokens is those in the same cluster as the one used for fine-tuning, and the second one is just the correct next tokens. For experiment efficiency, we only include the top-100 tokens before fine-tuning into the groups.

We illustrate the probability differences by NTP among instructions in ProbeSet in Figure 7, which shows in-cluster tokens to benefit more from fine-tuning than correct tokens. In fact, while the correct next tokens suffer from a $-0.99\%$ drop in probability sum on average, the in-cluster tokens reversely get a $+0.99\%$ lift in probability sum. This indicates learning correct knowledge does not naturally generalize to other correct knowledge, but with a high potential to generalize by spurious correlation in the key space. This discovery challenges the ability of language models to reflect the real world as they might generalize to hallucination injected by the spurious correlation in the key space.

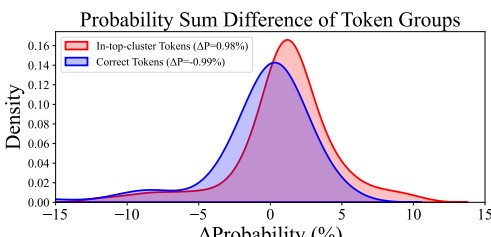

Figure 7: The distribution of probability switching of different next token groups in fine-tuning the language model.

Finally, we would like to propose some potential ways to further address the spurious correlation by editing the language modeling framework.

- **Adding a reranking stage**, which decodes top next tokens and uses a reranker to rescore them based on the context. Reranker is a commonly applied module in information retrieval and NTP can be viewed as a retrieval stage (query-key matching) from the view of information retrieval. The reranker scores with the predicted token in the context, allowing the next tokens to interact with the context to produce less biased NTP. A possible challenge is the ability of the reranker to recognize whether rather non-informative tokens (like subwords) can lead to a correct decoding path.
- **Adding a contextualization layer for vocabularies**, which adjusts the distribution of key vocabulary representations based on the context as an input. This strategy has potential as the query representations are well contextualized by the Transformer architecture, which can be extended to contextualize the key vocabularies. A potential challenge is the cost to contextualize the large scale vocabularies, which requires multiple times of interactions between them and the input context.

## 7 Conclusion and Future Work

In this paper, we unveil the potential spurious correlation in the key vocabulary spaces of neural language models for next token prediction. We use knowledge probing experiment to verify the potential issue and correspondingly propose in-context navigation for better token probing. We show in-context navigation can be extended to benefit open-ended and chain-of-thought generation. Finally, we discuss the further impact of the spurious correlation on language models and propose potential ways to address issues for future works.

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

## A    LIMITATION

The main limitations of our content in this paper are in the in-context navigation (ICN) query searching method, which is established on the ideal performance of large language models. Thus, it might not be applicable to weaker models, especially for language models that are not trained by supervised fine-tuning (e.g. GPT-2). The navigation performance of ICN might also be dependent on the generation ability of the language model itself as it might append incorrect results to the context, misleading the space for navigation. Finally, ICN requires encoding contexts to queries for multiple times, which will reduce the generation efficiency. In summary, ICN should be considered as a compromising method in the condition that the language model is frozen and thus can be fine-tuned. Some more fundamental ways to address the spurious correlation can be the ones we have discussed in training impact or changing the architecture of the language model to produce multiple queries. Besides further refinement in methods, our experiment scope can also be extended to other language modeling scenarios such as masked language modeling, which can be a potential future work.

## B    LLM DISCRIMINATOR ACCURACY VALIDATION

We validate the discriminative ability of `GPT-4o` by testing it on the dataset from CG-Expan (Zhang et al., 2020) with the prompts in Appendix K. The dataset include 10 categories, each with positive and negative examples. We showcase some examples in Table 7. `GPT-4o` achieves 92.71% accuracy on the test set, which

| Category | Positive | Negative |
|----------|----------|----------|
| Country | United Kingdom | London |
| Disease | Lymphoma | Chemotherapy |
| Party | Democrats | Episcopalians |

Table 7: Examples of data from CGExpan.

established it as a competent discriminator. As the entities generated from LLaMA might be different from the CGExpan test set, we manually check 10% (600 in all) of the discrimination result, which shows 94.67% accuracy. We find the accuracy to be higher because some entities generated from LLaMA makes no sense because of bad first-tokens. Thus, we conclude the discrimination of knowledge probing to be a easy task for `GPT-4o` to make trustful predictions.

## C    KNOWLEDGE PROBING CATEGORY LIST

| Sports League | Astronomical Objects | Scientist | Landmark |
|---------------|----------------------|-----------|----------|
| Basketball Sports League | Planet | Computer Scientist | European Landmark |
| Baseball Sports League | Nebulae | Social Scientist | Modern Landmark |
| USA Sports League | Fixed Stars | Female Scientist | Tower Landmark |
| European Sports League | Hot Astronomical Object | German Scientist | Natural Landmark |
| **Country** | **City** | **TV Channel** | **Restaurant** |
| Developing Country | Canadian City | USA TV Channel | Fast Food Restaurant |
| African Country | Coastal City | Entertainment TV Channel | French Restaurant |
| Small Country | Capital City | News TV Channel | Pizza Restaurant |
| Island Country | Asian City | Premium TV Channel | Expensive Restaurant |
| **Company** | **Creature** | **Disease** | **Food** |
| USA Company | Four-leg Creature | Infective Disease | Chinese Food |
| Japanese Company | Flying Creature | Stomach Disease | Spicy Food |
| Technology Company | Mammal | Childhood Disease | Veggie Food |
| Medical Company | Bacteria | Fatal Disease | Dessert |

Table 8: The full list of categories and sub-categories.

# D  PROBING RESULT ON STARTING CHARACTER

Figure 8: Probing result on generating words starting with given characters.

In Figure 8, we showcase a much simpler probing target than the main content - generating words starting with given characters, for instance, "P". We can easily evaluate the probing correctness by checking the first character from generation without any LLM. The observed result is consistent with the main content (except for "W" and "X") that tokens with high similarity to top tokens do not necessarily start with the same token, which introduces spurious correlation. This evaluation independent from LLM evaluator further validate the existence of the spurious correlation in NTP by key space similarity.

# E  RESULT WITH MULTIPLE SAMPLING AS APPROXIMATION

| Category Group | Sports League InTop | Sports League OutTop | Astronomical InTop | Astronomical OutTop | Scientist InTop | Scientist OutTop | Landmark InTop | Landmark OutTop | Country InTop | Country OutTop | City InTop | City OutTop |
|---|---|---|---|---|---|---|---|---|---|---|---|---|
| Accuracy | 41.95 | **54.42** | 40.75 | **63.27** | 38.80 | **46.91** | 51.34 | **68.06** | 41.07 | **61.90** | 55.62 | **67.45** |
| Rank | 54.01 | 56.97 | 54.42 | 56.65 | 52.70 | 58.51 | 54.79 | 56.10 | 54.69 | 55.71 | 52.00 | 59.29 |
| Proportion | 50.89 | 49.11 | 54.00 | 46.00 | 53.11 | 46.89 | 48.67 | 51.33 | 41.78 | 58.22 | 51.78 | 48.22 |

| Category Group | TV Channel InTop | TV Channel OutTop | Restaurant InTop | Restaurant OutTop | Company InTop | Company OutTop | Creature InTop | Creature OutTop | Disease InTop | Disease OutTop | Food InTop | Food OutTop |
|---|---|---|---|---|---|---|---|---|---|---|---|---|
| Accuracy | 50.97 | **59.07** | 62.50 | **74.70** | 68.75 | **74.05** | 58.36 | **74.48** | 52.51 | **68.50** | 56.66 | **68.96** |
| Rank | 54.59 | 56.96 | 52.03 | 59.39 | 52.04 | 59.82 | 50.54 | 61.69 | 50.50 | 59.64 | 50.07 | 59.87 |
| Proportion | 46.89 | 53.11 | 51.11 | 48.89 | 54.00 | 46.00 | 56.44 | 43.56 | 46.22 | 53.78 | 46.00 | 54.00 |

Table 9: Knowledge probing result with multiple sampling as approximation.

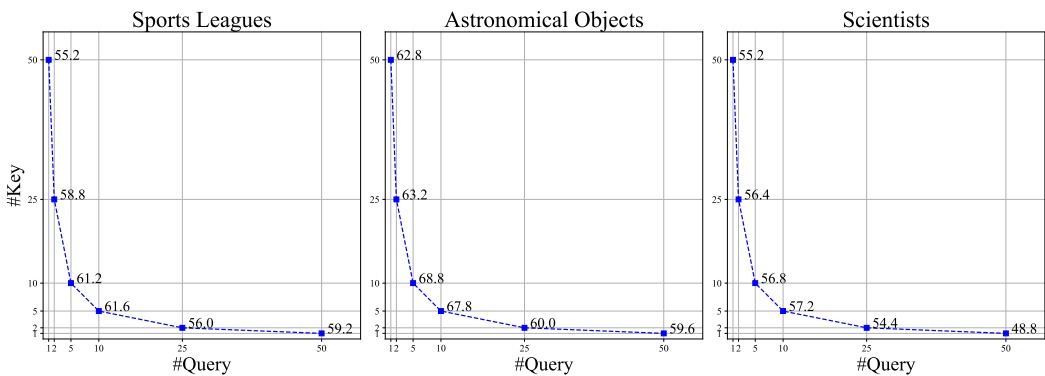

Figure 9: ICN results with multiple sampling as approximation.

In Table 9 and Figure 9, we represent knowledge probing result approximated by Monte-Carlo sampling (5 times each first-token), which shows a consistent result with the main content. For Figure 9, as MAP is inapplicable for multiple answers from a first-token, we report precision as the metric.

# F  EFFECT OF APPENDED EXAMPLES

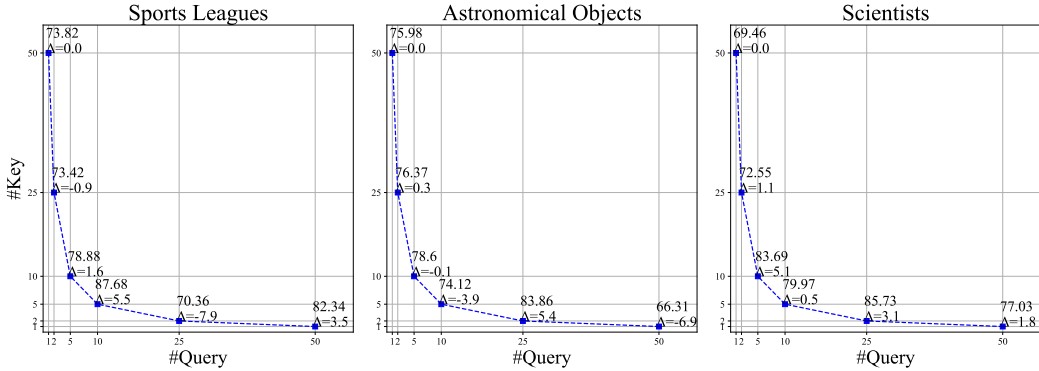

Figure 10: ICN results when probing after first-tokens with appended examples.

In the main content, the first-tokens are appended back to the initial prompt to avoid the influence of in-context examples on the generation space. In Figure 10, we show what if we append them in the prompt for probing. The result shows that in-context examples overall improve the probing precision. However, it also sometimes results in a significant drop when wrong answers are appended, which undermines the quality of in-context examples.

## G    RESULT ON OTHER LANGUAGE MODEL

| Category | Sports League | | Astronomical | | Scientist | | Landmark | | Country | | City | |
|---|---|---|---|---|---|---|---|---|---|---|---|---|
| Group | InTop | OutTop | InTop | OutTop | InTop | OutTop | InTop | OutTop | InTop | OutTop | InTop | OutTop |
| Accuracy | 39.92 | **47.72** | 42.18 | **56.50** | 23.21 | **35.64** | 47.69 | **47.99** | 42.95 | **43.53** | **64.12** | 61.14 |
| Rank | 59.67 | 53.34 | 53.79 | 57.51 | 53.25 | 61.64 | 54.90 | 56.74 | 54.25 | 58.27 | 52.94 | 60.10 |
| Proportion | 42.00 | 58.00 | 60.44 | 39.56 | 70.22 | 29.78 | 63.11 | 36.89 | 61.56 | 38.44 | 56.22 | 43.78 |

| Category | TV Channel | | Restaurant | | Company | | Creature | | Disease | | Food | |
|---|---|---|---|---|---|---|---|---|---|---|---|---|
| Group | InTop | OutTop | InTop | OutTop | InTop | OutTop | InTop | OutTop | InTop | OutTop | InTop | OutTop |
| Accuracy | 50.28 | **55.17** | 63.31 | 58.15 | 56.17 | **57.25** | 44.52 | **62.68** | 48.04 | **58.11** | 66.14 | **67.97** |
| Rank | 54.09 | 57.21 | 54.01 | 57.14 | 56.16 | 56.44 | 51.33 | 62.38 | 52.41 | 59.10 | 52.18 | 61.46 |
| Proportion | 46.00 | 54.00 | 61.33 | 38.67 | 47.56 | 52.44 | 62.44 | 37.56 | 55.11 | 44.89 | 70.22 | 29.78 |

Table 10: Knowledge probing results on `olmo-7b-instruct-hf`.

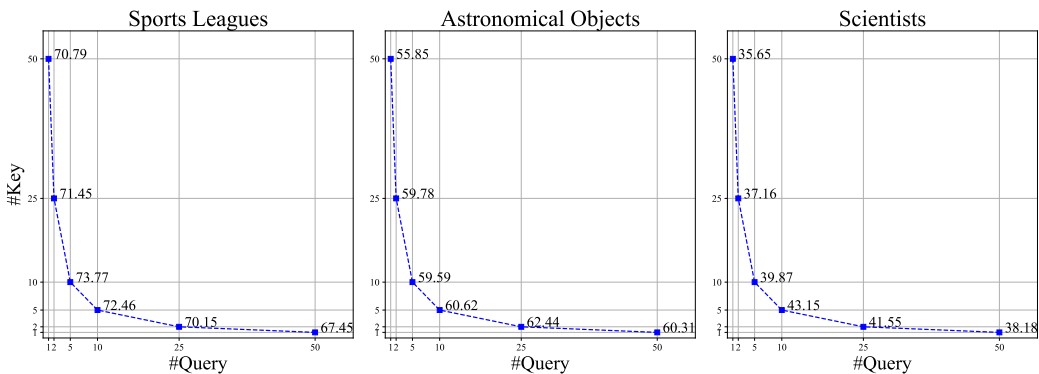

Figure 11: ICN results on `olmo-7b-instruct-hf`.

We include experiments on another open-source language model, `olmo-7b-instruct-hf`[1] (Groeneveld et al., 2024), which uses a different vocabulary dictionary from LLaMA. Our experiment shows result consistent to the main content in Table 10 and Figure 11. Thus, our discovery is more widely supported as a common issue among different language models.

## H    RESULT ON LARGER LANGUAGE MODEL

| Category | Sports League | | Astronomical | | Scientist | | Landmark | | Country | | City | |
|---|---|---|---|---|---|---|---|---|---|---|---|---|
| Group | InTop | OutTop | InTop | OutTop | InTop | OutTop | InTop | OutTop | InTop | OutTop | InTop | OutTop |
| Accuracy | 37.18 | **67.38** | 36.99 | **59.70** | 38.38 | **59.83** | 49.72 | **80.78** | 41.77 | **53.93** | 46.68 | **51.44** |
| Rank | 54.83 | 56.62 | 54.17 | 59.07 | 54.54 | 57.55 | 54.65 | 57.31 | 55.27 | 56.40 | 54.69 | 56.82 |
| Proportion | 64.22 | 35.78 | 69.11 | 30.89 | 71.11 | 28.89 | 62.89 | 37.11 | 58.22 | 41.78 | 58.89 | 41.11 |

| Category | TV Channel | | Restaurant | | Company | | Creature | | Disease | | Food | |
|---|---|---|---|---|---|---|---|---|---|---|---|---|
| Group | InTop | OutTop | InTop | OutTop | InTop | OutTop | InTop | OutTop | InTop | OutTop | InTop | OutTop |
| Accuracy | 39.39 | **58.02** | 51.96 | **69.51** | 69.73 | **75.52** | 55.45 | **67.55** | 50.43 | **76.61** | 48.33 | **81.08** |
| Rank | 57.17 | 53.51 | 53.30 | 60.35 | 53.7 | 59.72 | 53.94 | 59.03 | 54.96 | 56.09 | 53.91 | 58.66 |
| Proportion | 61.78 | 38.22 | 70.44 | 29.56 | 66.44 | 33.56 | 68.0 | 32.0 | 63.11 | 36.89 | 67.11 | 32.89 |

Table 11: Knowledge probing results on `llama-3-70b-instruct`.

In Table 11 and Figure 12, we illustrate the spurious correlation and ICN performance on the larger 70B LLaMA-3 model. We find a consistent conclusion with the discovery in the main content. Thus, the conclusion in our work is further scaled-up to even larger LMs.

---

[1]allenai/OLMo-7B-0724-Instruct-hf

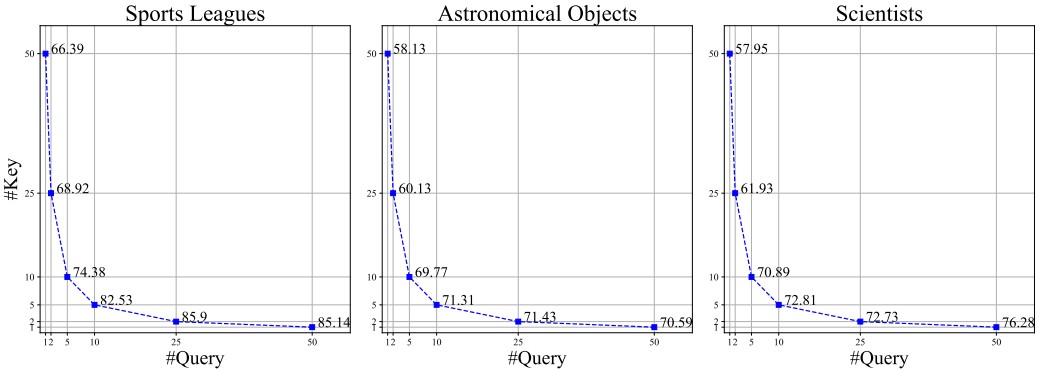

Figure 12: ICN results on `llama-3-70b-instruct`.

# I   RESULT ON DIFFERENT CLUSTERING RESULT

| Category | Sports League | | Astronomical | | Scientist | | Landmark | | Country | | City | |
| --- | --- | --- | --- | --- | --- | --- | --- | --- | --- | --- | --- | --- |
| Group | InTop | OutTop | InTop | OutTop | InTop | OutTop | InTop | OutTop | InTop | OutTop | InTop | OutTop |
| Accuracy | **57.08** | 55.78 | 51.95 | **75.82** | 53.13 | **65.52** | 70.78 | **76.68** | 64.11 | **70.84** | 64.37 | **76.86** |
| Rank | 52.76 | 57.61 | 54.27 | 58.13 | 54.57 | 56.90 | 55.79 | 54.46 | 54.39 | 55.61 | 52.77 | 61.25 |
| Proportion | 42.00 | 58.00 | 60.44 | 39.56 | 70.22 | 29.78 | 63.11 | 36.89 | 61.56 | 38.44 | 56.22 | 43.78 |

| Category | TV Channel | | Restaurant | | Company | | Creature | | Disease | | Food | |
| --- | --- | --- | --- | --- | --- | --- | --- | --- | --- | --- | --- | --- |
| Group | InTop | OutTop | InTop | OutTop | InTop | OutTop | InTop | OutTop | InTop | OutTop | InTop | OutTop |
| Accuracy | 61.03 | **67.77** | 68.53 | **82.24** | 74.78 | **80.72** | 71.32 | **78.25** | 65.84 | **72.94** | 72.24 | **74.91** |
| Rank | 54.69 | 55.88 | 54.02 | 58.51 | 52.84 | 58.47 | 53.13 | 61.07 | 49.47 | 58.3 | 51.88 | 58.30 |
| Proportion | 56.22 | 43.78 | 65.11 | 34.89 | 56.89 | 43.11 | 70.22 | 29.78 | 42.67 | 57.33 | 52.67 | 47.33 |

Table 12: Knowledge probing results on a different clustering result.

As the clustering algorithm might output different clusters by different initialization, we get another version of the clusters and rerun the experiment in Table 3. Table 12 presents our result, which is quite consistent with Table 3. Thus, our conclusion on the existence of spurious correlation is further solidified.

## J CASE STUDY

| Vanilla (#Query=1, #Key=50) | ICN (#Query=10, #Key=5) |
|---|---|
| Infectious diseases, Malaria, Ebola, HIV/AIDS, Tuberculosis | Infectious diseases, Malaria, Ebola, HIV/AIDS, Tuberculosis |
| The plague, Measles, Typhoid Fever, AIDS, Rabies | Pertussis, Measles, Cholera, SARS-CoV-2, Syphilis |
| Pertussis, SARS-CoV-2, TB, Smallpox, Bubonic Plague | Lassa Fever, Dengue Fever, Typhoid Fever, Rabies, Gonorrhea |
| Influenza, Plague, [OutLength], Covid-19, Avian Influenza | Herpes simplex, Salmonella, Trichomoniasis, Covid-19, Strep Throat |
| Ringworm, What is..., Cholera, Dengue Fever, MRSA | MRSA, Whooping Cough, Ringworm, Shigellosis, Flu |
| Mad Cow, Chickenpox, Tuberculosis, Swine flu, Strep Throat | Epidemic, Chickenpox, Campylobacter jejuni, Cryptococcosis, Plague |
| It's a..., Epidemic, Flu, Fever, Whooping Cough | Leptospirosis, Bubonic Plague, Q Fever, Scabies, Botulism |
| COVID-19, Aspergillosis, Common Cold, Lassa Fever, Toxoplasmosis | Clostridium difficile, Toxoplasmosis, Listeria monocytogenes, West Nile, Legionnaires' disease |
| Gonorrhea, You're going..., Worms, We have..., If you're... | Rocky Mountain, Psittacosis, [OutLength], Kuru, Aspergillosis |
| [OutLength], [OutLength], Monkeypox, Spanish Flu, Coronavirus | Histoplasmosis, Fever, AIDS, Nipah virus, Brucellosis |

Table 13: Case study of ICN in knowledge probing ("Infective Disease").

In Table 13, we showcase the effect of ICN on knowledge probing, which illustrates ICN to successfully navigate the query to correct first-tokens away from explored ones. Also, we can observe some nonsense generation introduced by spurious correlation in simply probing the top first-tokens.

| Vanilla (#Query=1, #Key=20) | ICN (#Query=4, #Key=5) |
|---|---|
| The company announced a significant increase in profits,... | The company announced a significant increase in profits,... |
| Apple's stock price surged 5% after the company... | Apple's stock price surged 5% after the company... |
| Amazon's stock price surged 10% after the company... | Amazon's stock price surged 10% after the company... |
| According to the latest report, the company's profits... | According to the latest report, the company's profits... |
| Tech giant Apple has announced a new line... | Tech giant Apple has announced a new line... |
| Microsoft's stock price surged 5% after the company... | Investors are optimistic about the company's new product... |
| Coca-Cola announced a major restructuring plan to cut... | In a major deal, tech giant Apple has... |
| E-commerce giant Amazon announced a 20% increase in... | A recent report by the International Monetary Fund... |
| Google's parent company, Alphabet, reported a 13% increase... | Microsoft's stock price surged 5% after the company... |
| Walmart has announced a major expansion of its... | Google's parent company, Alphabet, reported a 13% increase... |
| Alibaba's stock price surged 10% after the company... | Facebook's stock price plummeted 10% after the company... |
| Stock prices plummeted on Wall Street yesterday, with... | Tesla's stock price surged 10% after the company... |
| Global economic growth is expected to slow down... | Coca-Cola announced a major restructuring plan to cut... |
| Facebook's stock price plummeted 10% after the company... | China's economy grew at its slowest pace in... |
| Technology giant Apple has announced a significant increase... | Ford Motor Company announced a significant increase in... |
| Shares of XYZ Inc. plummeted 20% after the... | General Motors announced a major restructuring plan to... |
| Tesla's stock price surged 10% after the company... | Walmart has announced a major expansion of its... |
| Wall Street experienced a significant decline in stocks... | Goldman Sachs reports record profits for the quarter,... |
| After a successful IPO, the tech company's stock... | Procter & Gamble's quarterly profits rose 12% due... |
| Dow Jones Industrial Average (DJIA) closed at a... | Netflix's stock price surged 10% after the company... |

Table 14: Case study of ICN in open-ended generation ("Business News").

In Table 14, we showcase the effect of ICN on open-ended generation, which shows the first-token explored by ICN has better diversity, which also leads to more diverse sentence structure in generation.

| | |
|---|---|
| Question | Rita hand-picks Junebugs off of her plants every summer. On Monday, she removed 39 Junebugs.On both Tuesday and Wednesday, she removed twice as many Junebugs as she did on Monday. Thursday she removed 48 and on Friday she removed 57. What is the average number of Junebugs that she removes per day? |
| Answer | On both Tuesday and Wednesday, she removed twice as many as the 39 on Monday so she removed 2*39 = 78 Junebugs. All total she has removed 39+78+78+48+57 = 300 Junebugs She has removed 300 Junebugs over 5 days so she has removed, on average 300/5 = 60 Junebugs per day |

Table 15: The case used for chain-of-thought generation in Table 16.

| Vanilla (#Query=1, #Key=4) | ICN (#Query=4, #Key=1) |
|---|---|
| * **Step 1:**
Find the total number of Junebugs removed.
39 + 2 x 39 + 48 + 57 = 39 + 78 + 48 + 57 = 222
**Step 2:**
Find the average number of Junebugs removed per day.
To find the average, we divide the total number of Junebugs removed by the number of days.
222 / 5 = 44.4 | * **Step 1:**
Find the total number of Junebugs removed.
39 + 2 x 39 + 48 + 57 = 39 + 78 + 48 + 57 = 222
* **Step 2:**
Find the average number of Junebugs removed per day.
To find the average, we divide the total number of Junebugs removed by the number of days.
222 / 5 = 44.4 |
| * **Step 1:**
Calculate the total number of Junebugs removed:
39 + 2 * 39 + 48 + 57 = 39 + 78 + 48 + 57 = 222
**Step 2:**
Calculate the average number of Junebugs removed per day:
222 / 5 = 44.4 | * **Step 1:**
List the number of Junebugs removed each day:
Monday: 39
Tuesday: 2 x 39 = 78
Wednesday: 2 x 39 = 78
Thursday: 48
Friday: 57
* **Step 2:**
Add up the number of Junebugs removed each day:
39 + 78 + 78 + 48 + 57 = 300
* **Step 3:**
Find the average number of Junebugs removed per day:
300 / 5 = 60 |
| * **Step 1:**
We can add up all the Junebugs that Rita removed:
39 + 2 x 39 + 48 + 57 = 39 + 78 + 48 + 57 = 222
**Step 2:**
We can find the average by dividing the total number of Junebugs by the number of days:
222 / 5 = 44.4. | * **Step 1:**
Use the information that Rita removed twice as many Junebugs on Tuesday and Wednesday as she did on Monday.
This means that she removed 2 x 39 = 78 Junebugs on Tuesday and 78 Junebugs on Wednesday.
* **Step 2:**
Add up the number of Junebugs she removed each day:
39 + 78 + 78 + 48 + 57 = 300.
* **Step 3:**
To find the average, divide the total number of Junebugs by the number of days: 300 / 5 = 60. |
| * **Step 1:**
Calculate the total number of Junebugs removed:
39 + 2 x 39 + 48 + 57 = 39 + 78 + 48 + 57 = 222
**Step 2:**
Calculate the average number of Junebugs removed per day:
222 / 5 = 44.4 | * **Step 1:**
Rita removed 39 Junebugs on Monday.
* **Step 2:**
On Tuesday and Wednesday, she removed twice as many Junebugs as she did on Monday. So she removed 2 x 39 = 78 Junebugs on Tuesday and 78 Junebugs on Wednesday.
* **Step 3:**
On Thursday, she removed 48 Junebugs.
* **Step 4:**
On Friday, she removed 57 Junebugs.
* **Step 5:**
To find the average number of Junebugs she removed per day, we add up all the Junebugs she removed and divide by the number of days.
* **Step 6:**
The total number of Junebugs she removed is 39 + 78 + 78 + 48 + 57 = 300.
* **Step 7:**
The average number of Junebugs she removed per day is 300 / 5 = 60. |

Table 16: Case study of ICN in chain-of-thought generation.

In Table 6, we showcase how ICN helps the generation of diverse chain-of-thoughts, which consequently improves the performance in reasoning.

# K  PROMPTS, CASES AND HYPERPARAMETERS

In Table 17, we present the prompts used in our experiment for reproduction.

| Function | Prompt |
|----------|--------|
| Knowledge Probing | "Please show me some [CATEGORY]"
1. [ANSWER 1]
2. |
| Open-ended Generation | "Please show me some [CLASS] [DOMAIN]."
1. [SENTENCE 1]
2. |
| Chain-of-Thought | "[QUESTION] Please show me some different ways to solve this problem."
###Way 1: [CoT 1]
###Way 2: |
| Rephrasing | Reprahse this request: [REQUEST] |
| Probing Evaluation | "Please show me some [CATEGORY]. Is '[ANSWER]' considered as among the correct answers? Answer only 'Yes' or 'No'." |
| Get Answer (Numeric) | [CoT] **Final Answer (Only Number):** |
| Get Answer (Choice) | [CoT] **Final Answer (A, B, C, D, E):** |

Table 17: The prompts used in our experiments.

In Table 18, we showcase the rephrased knowledge probing requests generated by `GPT-4o` in the "Rephrasing" method.

| Original | Rephrased |
|----------|-----------|
| "Please show me some sports leagues." | "Could you please provide me with a list of sports leagues?" |
| "Please show me some USA companies." | "Could you provide me with a list of companies based in the USA?" |
| "Please show me some pizza restaurants." | "Could you provide information on some pizza restaurants?" |
| "Please show me some childhood diseases." | "Could you provide information on some illnesses that affect children?" |
| "Please show me some veggie foods." | "Could you please offer me a selection of vegetarian dishes?" |

Table 18: Cases in the rephrasing of knowledge probing requests.

For ZeroGen implementation, we fine-tune a RoBERTa Liu et al. (2019) (`RoBERTa-Large`) as the classifier, which is optimized by AdamW Loshchilov & Hutter (2019). The learning rate is initialized to $1 \times 10^{-5}$. The classifier is fine-tuned by 10 epochs with batch size 16. For the result, we report the averaged performance over 5 different runs.

