# OpenReview forum: "Correlation and Navigation in the Vocabulary Key Representation Space of Language Models"
_ICLR.cc/2025/Conference — ICLR 2025 Poster_

### Official Review · Reviewer_Re3A · 2024-10-31

**Soundness:** 3
**Presentation:** 3
**Contribution:** 3
**Rating:** 8
**Confidence:** 3

**Summary:**

Following the terminology in the paper, let us denote the encoded input hidden sate representation at some position (the one that is multiplied with the vocabulary matrix to get the probability distribution at that position) as "query", and let us denote the vectors in the vocabulary matrix (each correspondent to some token in the vocabulary) as "keys".

The paper explores the confounding factor that is brought about by the input-context-insensitive similarity of keys in their vector space in next token predictions (NTPs). Particularly, in the context where there are multiple good answers for an input prompt, many of the good answers are not as highly ranked, because input-irrelevant vocabulary keys can get highly ranked because they are close (in the non-contextual vocabulary embedding space) to some of the top ranked keys that are actually relevant to the input. Essentially, even if the top ranked keys are appropriate, they being scored high leads to a side effect of inappropriate (unrelated to input) keys to be also ranked moderately high only because they are close to the top keys in the vocabulary embedding space (which is insensitive to input context by design).

1. The paper provides empirical analysis of this phenomenon by probing LLMs.
2. The paper explores several ways to mitigate this issue - example by reranking/rephrasing and the main proposed method ICN (in-context navigation). ICN - employees iterative prompting to sequentially generate multiple good answer candidates (instead of just multi-sampling), where the model is prompted explicitly to generate something other than prior generated answers -- thus allowing to focus on different vocabulary key clusters unrelated to the prior answer-clusters.
3. The paper presents the utility of ICN as a strategy in different contexts - like incorporating it with self-consistency for mathematical reasoning or investigating open-ended generation.

**Strengths:**

1. Provides a reasonable analysis of an interesting phenomenon in the context of LLMs. Devises reasonable ways to quantify it.
1. Proposes a simple method (ICN) to mitigate the identified issue, and shows its utility empirically in a wide set of experimental contexts.
1. In the end, suggestions some interesting ideas to potentially solve this issue in a more principled manner (contextual layer for vocabulary or re-ranking).
1. The paper is well written (lower presentation score because of weakness in literature review)

**Weaknesses:**

1. Similar analyses have been already done. If I understand correctly, this issue is related to the well-known softmax-bottleneck phenomenon. And there are a bunch of related works on that - especially see [1]. While I think the current paper has some virtues over some of those prior works (like investigation in terms of LLMs, exploration of a tuning-free method ICN to mitigate it, and potentially better empirical measure by cluster divisions and such of this issue) - the past related works should be still acknowledged and used for contextualization.

1. If I am not missing anything, most of the analyses and investigation is based on a single LLM (Llama 3 8B instruct). While olmo results seem to be presented in the appendix, but ICN-related experiments seem to be missing for olmo. So the universality of the utility of ICN is still a question.

1. There could be more discussion about how much does this issue have anything to do with NTP in particular. Seems like a general issue -- that can apply to other language modeling settings (like Masked language modeling).

1. The technical details hyperparameters and the prompt settings are not crystal clear to me.






[1] Softmax Bottleneck Makes Language Models Unable to Represent Multi-mode Word Distributions - Chang et al. ACL 2022
 https://aclanthology.org/2022.acl-long.554/

**Questions:**

1. Line 256-257 is phrased in a confusing manner (and probably wrongly). Instead of "specifically, from
top-N (N = 100) to top-(K + 1) (K = 10)", you should probably say something like "specifically, from
all tokens in top-N (N = 100) but not in top-K (K = 10)".

1. I think the equation in 261 defining InTop is wrong. It should have $j \leq K$ not $K \leq j$. j is supposed to be one of the top-k tokens (thus $\leq K$) based on the text description.

1. Related to the last weakness: I think it would be good to more explicitly show what prompt template is used for ICN in different contexts and maybe some concrete examples about how iteratively the input prompt is being changed for different cases (mathematical reasoning, open ended generation, knowledge probing). I saw the case studies in the appendix but they seem to focus only on the output, or just the original input and output (rather than the processed prompt input with all the templates and such). And Table 16 doesn't seem to show anything related to ICN.

1. Still somewhat related to the last weakness: I am a bit confused by your definition of COT prompt in Table 16. Normally, COT prompt is supposed to be add something like  "Let's think step by step" to the beginning of the assistant response. Your COT prompt seems to be just about generating multiple answers. Moreover, self-consistency normally used distinct samples, whereas this COT prompt seems to be designed to generate multiple answer procedures in a single sample.

1. What exactly is Top in Table 6 or 5? Do you mean greedy decoding? If so I am not sure how that can be used to generate different sequences for diversity measures or self-consistency voting.

---

> ### Author Response · Authors · 2024-11-20
> **Author Rebuttal**
>
> We are thankful for your positive rating and valuable suggestions, which are applied to the rebuttal version to further solidify our work. We hope our rebuttal below can help address your concerns about our work.
>
> **Relation to previous works**
>
> Indeed, our work is related to the softmax issue, and we have updated the rebuttal version by citing your recommended paper to discuss our relation to it. Specifically, besides your mentioned development in the softmax issue, our work also extends the scope to diversity (Section 5) and fine-tuning  (Section 6) limitation, which further signifies our contributions to the softmax bottleneck. We have also discussed a pile of works about the softmax bottleneck to better position our work in the development of this field. We hope this will further clarify our paper’s background and contribution to readers.
>
> **ICN experiments on other LMs**
>
> We have already included ICN experiment results for olmo in **Figure 11**, which shows the ability of ICN to navigate the query for different LMs (over simple sampling when #Query=1). Furthermore, we have added experiments on spurious correlation and ICN of a much larger LM “meta-llama/Meta-Llama-3-70B-Instruct”, which is presented in **Appendix H**, showing consistent results on very large LMs with other experiments.
>
> **The generality of spurious correlation**
>
> We definitely agree with the existence of a spurious correlation issue among more LM learning scenarios such as masked language modeling. We have added the discussion of a broader application to Appendix A, which can help readers further understand the potential extension of our work in future works.
>
> **Details in implementation**
>
> We have updated the prompts and templates in Appendix K for the three main tasks in our paper for better clarity. We have also attached corresponding prompting cases to better illustrate the workflow of the ICN technique. For the CoT prompt, we do not explicitly mention “Let’s think step by step” because LLMs generally search for CoTs when prompted to solve math problems. Thus, we just prompt the LLM to propose different ways (which implicates CoTs) to solve the math problems.
>
> **Clarification of the abbreviation in Tables**
>
> “Top” in the Tables refers to probing the first tokens ranked above a certain threshold as mentioned in the second paragraph of Section 5.3. More specifically, we append top-K first tokens predicted to the prompt and then use greedy decoding to complete the rest of the generation. We have updated the caption in Tables 5 and 6 to further clarify the abbreviation for readers.
>
> **Typos**
>
> We have corrected the formulas to make them consistent with our definitions in the rebuttal version. Thank you very much for pointing them out!

---

> > ### Comment · Reviewer_Re3A · 2024-11-20
> >
> > Thank you. Most of my concerns are addressed. I increased the score.

---

> ### Author Response · Authors · 2024-12-02
>
> Dear Reviewer Re3A,
>
> We enjoyed the discussion with you during the rebuttal period, which is valuable for strengthening our paper's connection with previous works. We appreciate your clear positive attitude toward our paper and feel glad that your concerns have been addressed by our response. Thanks for your time and thoughts devoted to the discussion!

---

### Official Review · Reviewer_7cDM · 2024-11-03

**Soundness:** 4
**Presentation:** 3
**Contribution:** 4
**Rating:** 8
**Confidence:** 4

**Summary:**

The paper discover a fundamental issue that the next-token prediction is affected by correlations within the vocabulary key space. By analyzing key representations in transformer LMs, the authors observe that context-agnostic similarities between certain keys can lead to spurious correlations, such as the capital alphabets are predicted with high probabilities together without regarding their real relevance. These correlations cause certain “middle-ranked” tokens, often unrelated semantically to the top-ranked token, to be underestimated in ranking. This issue introduces bias and reduces sampling diversity in open-ended or exploratory text generation. The author further proposed In-Context Navigation (ICN) to incorporate the previous generated outputs into the input context to iteratively guide the LM away from explored key regions.

**Strengths:**

- Novel discovery on the the issue of very fundamental design of transformer LMs. It's valuable to facilitate the understanding of LM decoding biases
- Novelty in proposed approach ICN to mitigate such bias.
- The paper shows strong empirical evidence through diverse tasks, including knowledge probing and chain-of-thought generation, to validate ICN’s effectiveness.
- The author proposed some potential direction to solve this issue in the LM pretraining stage, which can be positive contributing to the community.

**Weaknesses:**

- Lack of exploration on larger models: the paper only discuss LLaMA-3-8B in the experiments. However, I expect that a larger model with higher-dimensional embedding space can more or less mitigate such issue. When the dimension is large enough, the query embedding can be similar to different key embeddings in many different sub-spaces, so as to reduce the problem of similar key vectors always get activated together. It will be good to show more results in the larger models like LLaMA-3-70B.
- ICN increase the context length. When making the model generating longer responses (e.g. CoT) while considering previous generated outputs, ICN can largely increase the context length and inference cost.

**Questions:**

- I wonder why a regular decoding process cannot achieve the same effects of ICN did? In regular decoding, the previous generated outputs are natively placed in the input context. If we put always put the generated outputs in the context, we should be able to achieve the same thing as ICN did, right?
- Is there any analysis of how ICN can increase the context length and inference cost?

---

> ### Author Response · Authors · 2024-11-20
> **Author Rebuttal**
>
> We sincerely appreciate your strong recommendation for our work, thank you! We have made several beneficial updates to our rebuttal version and are willing to address your concerns with the rebuttal below.
>
> **Larger model experiments**
> We followed your suggestions to investigate the results on larger models and our results have been attached to **Appendix H**. The experimented “meta-llama/Meta-Llama-3-70B-Instruct” model has an 8192 embedding size, which is twice as large as the 4096 embedding size of “meta-llama/Meta-Llama-3-8B-Instruct”. However, the spurious correlation still significantly exists, which indicates this issue is not mitigated by simply scaling up the model size (or embedding dimension). Moreover, our ICN shows consistent improvement on the 70B model, which supports the effectiveness of ICN for larger models.
>
> **ICN efficiency**
> ICN does have efficiency limitations as we have discussed in Appendix A. This limitation emerges from the necessity of searching for multiple queries to traverse across different clusters. Our experiments also include the trial to navigate out of clusters by reranking with cluster information, which shows some improvements with little efficiency burden. However, searching for new queries can lead to much more significant improvements, which motivates us to propose ICN as our main solution by trading off some efficiency.
>
> **Comparison with regular decoding**
>
> In regular decoding, the LM is decoded to get one answer each time, which is automatically appended to the context. This scenario is a special case in ICN when #Key = 1, as discussed in lines 334-338. Our discovery reveals that #Key = 1 generally does not lead to the best performance in comparison with a balance between #Query and #Key, which is a conclusion shown in Figure 5.

---

> > ### Comment · Reviewer_7cDM · 2024-11-20
> >
> > Thanks for the reply! The response addressed my concerns. It's interesting to see the large models still have this issue. I will keep my scores to accept this paper!

---

> ### Author Response · Authors · 2024-12-02
>
> Dear Reviewer 7cDM,
>
> We enjoyed the discussion with you during the rebuttal period, which is valuable for extending our conclusion to larger LMs. We appreciate your clear positive attitude toward our paper and feel glad that your concerns have been addressed by our response. Thanks for your time and thoughts devoted to the discussion!

---

### Official Review · Reviewer_2odz · 2024-11-05

**Soundness:** 2
**Presentation:** 3
**Contribution:** 2
**Rating:** 5
**Confidence:** 4

**Summary:**

This paper present an interesting fact that the vocabulary representations (keys) has spurious correlations with the generation process. More specifically, when some token gets a higher generation probability, the tokens that are closer to this token will also have higher generation probability, even though they may not lead to correct generation results.

The paper also present a simple solution, which includes the explored decoding results in the context and prompt the LLM to generate differently. Experiments on knowledge-probing tasks show that the method leads to exploration away from the explored keys.

**Strengths:**

The paper use clustering and visualization to show that the similarities in keys did resulting in generation bias.

The proposed method, ICN, could increase the generation probability for tokens that are different from previous explored keys.

**Weaknesses:**

I dont think it is reasonable to ask the LLM to be correct for the top 100 candidate. At least it has already shown strong performance on the top 10 or maybe to the top 20 candidate.

I think ask the LLMs to generate different tokens is an practical approach, but it will also decreases the generation probability for correct answers that shows the same tokens with those explored ones.

**Questions:**

Since the key representations are learned during large scale pretraining, it is also possible that the generation probability behind these keys still matches the knowledge in the world. As shown in Figure 2, there are quite a lot of correct key around the query. It is not possible, in this case, to have all the correct answer appears around the query anyway.

I am wondering what prompt is used for rephrasing and reranking, respectively. I am wondering why rephrasing does not bring to much difference.

I am also curious about how the method related to the contrastive decoding approach.

[Contrastive Decoding: Open-ended Text Generation as Optimization. ACL 2023]

---

> ### Author Response · Authors · 2024-11-20
> **Author Rebuttal**
>
> We are sincerely thankful for your constructive suggestions and questions about our work! We have updated our rebuttal version of the paper with the help of your suggestions and would like to address your concerns in the rebuttal below.
>
> **Experiment design**
>
> The main aim of our knowledge-probing experiment is not to ask for the high correctness of the top 100 candidates but to verify the existence of a spurious correlation between vocabulary embeddings. Thus, we divide the middle-ranked tokens (top 11-100) into two balanced groups, one has higher ranks but lower accuracy and the other has lower ranks but higher accuracy. As the former group of tokens (with higher ranks but lower accuracy) is in the same cluster as those top-ranked ones, we conclude the existence of the spurious correlation that overestimates the ranks of tokens near the top predictions.
>
> **Sampling with explored tokens**
>
> We sample multiple times with the start of the same token, which is included in the experiments in Table 9 and Figure 9 in Appendix E. The experiment results show we can sample answers starting with the same tokens more accurately with the application of ICN. In other experiments, we use greedy decoding to approximate the correctness of the next token prediction (for experiment efficiency), which thus does not include other correct answers starting with the explored tokens.
>
> **The distribution of correct/incorrect keys**
>
> Our results show the limitation of LMs in reflecting the real-world knowledge distribution. We find the next token distribution is highly controlled by the vocabulary key embeddings. Considering the case in Figure 1, “Skunk” and “Racoon” are well-known “black and white animals”, but are ranked behind answers starting with “Q”, which hardly contains “black and white animals”. Thus, the spurious correlation between “Q” and the popular next token “P” (-> “Panda”) hurts LMs’ reflection of the real world. Moreover, the unlikeliness of having all correct answers around the query is an important discovery of our paper, which challenges the memorization ability of LMs expecting LMs to rank frequent knowledge (“Skunk” and “Racoon”) higher than rare ones (answers starting with “Q”).
>
> **Prompts in experiments**
>
> We have attached the rephrasing prompts and some rephrasing cases to Appendix K. Based on Figure 4, we attribute the limited rephrasing difference to LLM encoding semantically similar prompts to similar positions, which is further derived from the instruction-following requirement of LLMs to perform similarly on semantically similar prompts. In contrast, our ICN injects extra semantics (generation constraints) to the prompt, which can thus effectively find new queries with lower similarity with explored queries. Reranking does not require a prompt but rather uses cluster information to adjust the initial rank, thus there is no prompt for reranking to be included in the appendix.
>
> **Relation to contrastive decoding**
>
> Our work has a connection to contrastive decoding as both works use negative examples (Generation by weaker LMs in contrastive decoding and explored generation in our case) to improve the generation quality. The main difference is the layer of LMs we are focusing on. Contrastive decoding focuses more on processing the probability produced after the LM head, which utilizes the gap between small and large models in producing correct predictions. Our work focuses more on the distribution of queries and keys before the LM head and discusses how the key distribution leads to unexpected outputs from the LM head caused by spurious correlation. We have cited and mentioned this method to be another trial to exploit NTP predictions for LM decoding in the related work section.

---

> > ### Comment · Reviewer_2odz · 2024-11-20
> >
> > Thanks for the explanation. I agree the spurious correlation between vocabulary embeddings and the output is an interesting finding of this paper. But I still have doubt about the ICN approach, I think it is actually a trade-off, between increasing the probability of long-tail results and retaining the probability distribution learnt during pretraining.

---

> > > ### Author Response · Authors · 2024-11-21
> > > **Author Rebuttal - Initial probability might not ideally reflect full knowledge, while ICN can help address this issue**
> > >
> > > Thanks for your further question on the effect of ICN! The following is our corresponding response:
> > >
> > > - **Spurious correlation harms probability in reflecting knowledge learned during pretraining**
> > >
> > > Because of the spurious correlation, the learned probability distribution will always put tokens similar to top predictions high in the ranking. For instance, if an LM predicts "P", the LM will be more likely to predict "A"-"Z", and consider many correct tokens after that, which does not reflect an ideal distribution of the learned knowledge during the pre-training. This phenomenon is verified by the experiment in the **current section 6**, that learning new knowledge (such as "P") generalizes more to spuriously correlated tokens (such as "A"-"Z") rather than other correct tokens. Therefore, due to the spurious correlation, LM cannot reflect the real-world distribution by its predicted probability distribution.
> > >
> > >
> > > - **Long-tailed knowledge in probability might not be long-tailed in real distribution**
> > >
> > > Considering an extreme case we have 900 sentences "X is red" and 100 sentences "X is round" during the pretraining. Ideally, we will get the prediction on "X is" to be 90% "red" and 10% "round" (other words with very low probability). However, there is a cluster of color words (including "blue", "green", etc.) and the query of "X is" will be pulled toward other words in the color word cluster. This will significantly reduce the probability of "round" and might even put some color words such as "blue" and "green" over the real learned knowledge "round" in the ranking, even if "round" has been learned multiple times during pretraining.
> > >
> > > - **ICN gives LM a chance to generate more comprehensive knowledge learned during pretraining**
> > >
> > > Our ICN indicates that while "round" cannot be well reflected by prompting "X is", there is a chance by prompting "X is red and" to retrieve the knowledge "round" when "red" probability is eliminated by explicit context. Thus, we agree there to be a trade-off, but the trade-off is between **"diving deeper to the current cluster affected by spurious correlation"** or **"jumping to knowledge stored in other clusters not reflected by the limitation of probability"**. The latter one potentially better comprehends the real-world knowledge distribution skewed by spurious correlation.
> > >
> > > - **A case study on whether the knowledge discovered by ICN is simply long-tailed knowledge or victim of probability limitation in modeling across clusters.**
> > >
> > > | Iteration| 1  	| 2 	| 3 	| 4 	| 5 	| 6  	| 7  	| 8  	| 9  	| 10 	|
> > > |-------------|--------|-------|-------|-------|-------|--------|--------|--------|--------|--------|
> > > | Avg. Rank| 3.0	| 14.2  | 11.6  | 40.4  | 14.8  | 206.6  | 98.4   | 127.6  | 858.2  | 228.4  |
> > > | Precision	| 1.0	| 1.0   | 1.0   | 0.6   | 1.0   | 1.0	| 1.0	| 0.8	| 1.0	| 0.6	|
> > > We run an ICN for "sports leagues" with #Query = 10 (Re-encode new queries 9 times) and #Key = 5 (Probe top-5 tokens each time).
> > >
> > > **Avg. Rank** refers to the average rank in the initial probability distribution of the 5 tokens predicted in this iteration.
> > >
> > > **Precision** refers to the precision of the 5 tokens predicted in this iteration.
> > >
> > > From the results, it becomes evident that it is overly simplistic to classify the tokens identified by ICN as simply long-tailed. For instance, in iteration 9, the five tokens have an average rank of 858.2 in the initial probability distribution, yet all of them are correct. This suggests that these tokens represent knowledge clusters - closely aligned with the query in iteration 9 - that were overlooked due to spurious correlations in the initial prompt, rather than purely statistical long-tailed knowledge.

---

> ### Author Response · Authors · 2024-11-24
>
> Dear reviewer 2odz, we sincerely appreciate your valuable feedback and have made the necessary clarifications and updates based on your suggestions. Thank you for your efforts in helping us further refine our work, especially in the working mechanism behind ICN!
>
> As the discussion period approaches its conclusion, we kindly ask if you could spare some time to review our rebuttal and evaluate whether it addresses your concerns. We would be delighted if you find our responses satisfactory and consider adjusting your rating accordingly. If there are any remaining questions or concerns, we remain actively available to address them promptly.
>
> Thank you once again for your time and thoughtful input!

---

> > ### Author Response · Authors · 2024-11-26
> >
> > Good news! The discussion period has been extended by 6 days. Please don't hesitate to express your potential confusion about the details of our work and response. We are looking forward to more valuable feedback from you. We will stay passionate to address them for you, which will also benefit our work. Thanks for the time and effort you spend on our work!

---

> ### Author Response · Authors · 2024-12-02
>
> Dear Reviewer 2odz,
>
> We enjoyed the discussion with you during the rebuttal period, which is valuable for clarification of the method in our paper. As the discussion period finally approaches its end, we hope our latest response has fully addressed your remaining concern about the effect of ICN. If you have any further questions, please feel free to share them with us! In the last two days, we will continue showing our greatest respect for your devotion to the discussion by staying active for any questions. Thanks again for your time and thoughts!

---

### Official Review · Reviewer_ciXY · 2024-11-09

**Soundness:** 3
**Presentation:** 2
**Contribution:** 2
**Rating:** 5
**Confidence:** 4

**Summary:**

This paper investigates the impact of word-embeddings parameters in the last layer (also referred to as un-embedding parameters, referred to as "keys" in this work) on spurious correlations during decoding via language models. Specifically, this paper finds that top-k items in response to a prompt tend to be close together (low cosine distance) and hence a lot of plausible and correct responses which might have a higher cosine distance from the top-k items are ranked lower in LM predictions. This impact is shown on knowledge probing tasks which shows that a significant number of correct answers lie outside of the cluster formed by top few items. To ameliorate this issue, the paper proposes a simple prompt-based fix called ICN, which appends the top-few items to the query in a listed form to encourage the language model to suggest "under-explored" items. This approach is shown to improve accuracy on knowledge probing. This approach is also used for open-ended generation to stimulate diversity among the responses. When this approach is used for CoT consistency decoding, it tends to improve upon standard CoT sampling baseline slightly. Finally, this paper also provides evidence that fine-tuning doesn't change the unembedding layer much and is still susceptible to the same spurious correlations as pre-fintuned models.

**Strengths:**

-- This paper is well-motivated and identifies a focused problem in natural language generation. However, as I note in the weaknesses section, it misses out on discussion about other work investigating the inadequacy of softmax for similar reasons this paper explores.

-- The effects shown are significant. The knowledge probing experiments show concerning effects of the spurious correlations introduced due to the unembedding parameters.

-- The proposed fix, while simple seems to be effective at ameliorating the issue.

-- The proposed approach also seems to improve diversity of generated text.

**Weaknesses:**

--  The paper should improve the presentation. I am unsure about the exact prompt behind ICN while I understand it at a high level. Moreover, I it is also difficult to understand how this prompting scheme was explicitly adapted for open-ended generation and CoT. Clear examples would help. Also, the baselines should be clearly explained in greater detail.

-- The proposed ICN scheme also seems expensive to implement as a general method. The results indicate that one needs a large number of queries and a large number of keys to work.

-- My biggest concern is that this work ignores a lot of prior work that precisely characterizes the problem of spurious correlations due to the unembedding parameters. [1] introduces the popular concept of softmax bottleneck which shows rank-deficient behavior of the softmax function during decoding which is directly linked to the unembedding parameters. [2] investigates exactly the problem this paper studies. [3] relatedly shows that some words might never be chosen as argmax even if correct because of the manifold of the unembedding parameters.

[1] Breaking the softmax bottleneck: A high-rank RNN language model. Yang et al. 2017

[2] Stolen probability: A structural weakness of neural language models. Demeter et al. 2020

[3] Low-Rank Softmax Can Have Unargmaxable Classes in Theory but Rarely in Practice. Grivas et al. 2022

All of these works and more are closely related to the proposed work and should be discussed in detail.

-- It is not discussed/analyzed in detail what makes ICN work. The prompt as designed could have worked against the desires as well in which it explores the nearest items more eagerly. I don't come away with a good understanding of why this method of prompting is effective over other potential prompts.

**Questions:**

Please address the concerns above.

---

> ### Author Response · Authors · 2024-11-20
> **Author's Rebuttal - Position of our work in un-embedding parameters analysis**
>
> We are greatly thankful for your mentioned related works on un-embedding parameters (key representations)! Our paper emphasizes more on sampling precision in the diverse generation, which misses the related works on the issues caused by the context-agnostic un-embedding parameters. We have cited and discussed our work's relationship to these studies in the **introduction and related work sections** of the current rebuttal version. Additionally, we are happy to provide a concise explanation of how our work differs from these previous studies here.
>
> **Related Work 1**: “Stolen probability: A structural weakness of neural language models.” (Demeter et al. 2020) As you mentioned, this work has the most similar topic to us, which shows a next token's probability (especially with a low-norm vector) can be stolen (and thus bounded) by other tokens, when it falls inside a convex hull.
>
> **Difference:** The essential difference between our work and  (Demeter et al. 2020) is the victim of probability stealing and whether other tokens’ correctness is taken into account for distributional precision. (Demeter et al. 2020) concentrates on the top-ranked tokens whose maximal probability is bounded by other tokens. Our work concentrates on the correct middle-ranked tokens, whose ranks are lower than incorrect tokens near top predictions (spuriously correlated to top tokens). **With only (Demeter et al. 2020), one may still wonder whether probability stealing is somehow beneficial to generation diversity by sharing probability with nearby similar tokens.** Our work points out the spurious correlation between embedding similarity and correctness (nearby tokens can be incorrect and faraway tokens can be correct because they are in different clusters), which extends the discussion of probability stealing from top-token probability to distributional precision. Our discovery even challenges the LLM’s memorization ability in Section 6 as it tends to memorize the next tokens by groups (clusters), which includes many noisy tokens.
>
>
> **Related Work 2:** “Low-Rank Softmax Can Have Unargmaxable Classes in Theory but Rarely in Practice.” (Grivas et al. 2022) is a following work of (Demeter et al. 2020), which shows probability stealing can cause certain classes (next tokens) never appear as the top prediction when the bias term is added.
>
> **Difference:** Our work has a difference in the studied subjects (middle-ranked tokens and distributional precision) with (Grivas et al. 2022), as mentioned in the difference with (Demeter et al. 2020).
>
> **Related Work 3:** “Breaking the softmax bottleneck: A high-rank RNN language model.” (Yang et al. 2017) is a pioneering study on the limitation of un-embedding parameters but is not so strongly related to our topic as the two works above.
>
> **Difference:** As mentioned in Section 4.3 of (Demeter et al. 2020), softmax bottleneck (Yang et al. 2017) concentrates more on the rank difference between the high-rank natural language and the low-rank un-embedding parameters. Our work and the two related works above concentrate on issues caused by the arrangement of vocabulary embeddings.
>
> In summary, our work concentrates on the distributional inaccuracy caused by ranking incorrect nearby tokens over correct faraway tokens (middle-ranked tokens as victims), differing from the concentration on probability bounds in previous works. For softmax bottleneck (Yang et al. 2017), our work is distinct in concentrating on the arrangement of the vocabulary embeddings. These differences have been carefully added to the rebuttal version, which strengthens our connection with previous works and further clarifies our contributions in the development of un-embedding parameter analysis (from simple top-token probability to more complex distributional precision).

---

> > ### Author Response · Authors · 2024-11-20
> > **Author's Rebuttal - Position of our work in un-embedding parameters analysis (Rebuttal Revision)**
> >
> > We have made the revisions shown below,
> >
> > **Introduction**
> > > However, previous works have shown NTP distributions are highly restricted by the arrangement of key embeddings in a context-agnostic manner, such as probability bounds for less frequent words **(Demeter et al., 2020)** (unpredictable in extreme situations **(Grivas et al., 2022)**). With the discovery of context-agnostic embedding's limitation in modeling multi-mode word distributions **(Chang & McCallum, 2022)**, we are motivated to explore and mitigate the potential spurious correlation in the embedding space. Different from the major focus on probability bounds in previous works, we emphasize more on distributional precision - is NTP rank a good reference for diverse sampling?
> >
> > **Related Work**
> > > Unfortunately, LMs might not be able to model such diverse decoding paths because of the softmax bottleneck **(Yang et al., 2018)** as they are inclined to monomodal words **(Chang & McCallum, 2022)**. The probabilities of less frequent words are also likely to be bounded when falling inside convex hulls in the distribution **(Demeter et al., 2020)**, causing some words to be unpredictable **(Grivas et al., 2022)**. Based on previous efforts, our work further illustrates and mitigates the potential spurious correlation phenomenon in NTP.

---

> ### Author Response · Authors · 2024-11-20
> **ICN efficiency, Presentation detailedness, and Working mechanism behind ICN**
>
> **ICN efficiency**
>
> ICN does have efficiency limitations as we have discussed in the limitation section (Appendix A). This limitation emerges from the necessity of searching for multiple queries to traverse across different clusters. Our experiments also include the trial to navigate out of clusters by reranking with cluster information, which shows some improvements with little efficiency burden - referring to the **“Rerank”** performance in Table 4. On the other hand, we find searching for new queries can lead to much more significant improvements, which motivates us to propose ICN as our main solution by trading off some efficiency.
>
> **Presentation detailedness**
>
> We have attached ICN templates and corresponding cases for open-ended generation and chain-of-thoughts generation to Appendix K. For baseline descriptions, we have updated the paragraph describing baselines in Section 5.2 with more details on high-level aims and specific implementations. Our updates can help the reader form a clearer understanding of the methods in the experiment comparison.
>
> **Working mechanism behind ICN**
>
> We understand the potential concern that the explored keys appended to the context might serve as in-context examples to encourage LMs to explore nearby tokens harder. **However, this is avoided in the performance of ICN because the instruction indicates a very low probability of explored keys (very large query-key distance).** Thus, those tokens near explored keys will also be far away from the new query encoded by ICN. The key hint here is the correlation between keys, which can also be used to navigate queries away from explored keys as well as their nearby tokens. The design of ICN prompts is also limited to the case shown in our paper. There might be other prompt implementations showing better navigation performance in ICN and our prompt showcases the applicability of ICN’s high-level concept.

---

> > ### Comment · Reviewer_ciXY · 2024-11-26
> > **Thanks for the rebuttal**
> >
> > Hello, thanks for the rebuttal and contextualizing your work in prior work and improving the presentation. My overall impression remains the same as earlier in light of prior work and the approach's efficiency. However, I have increased my score to 5 because of the changes made.

---

> > > ### Author Response · Authors · 2024-11-27
> > > **Further clarification - Efficiency and Prior Work**
> > >
> > > Good to further discuss with you! We sincerely appreciate your new feedback on the ICN efficiency and prior work. We are glad to clarify the mentioned points to address your concerns.
> > >
> > > **ICN Efficiency**
> > >
> > > We want to clarify that ICN introduces only a minimal computational overhead to the probing process. The LLM forwarding is employed in two stages: first-token probability prediction and answer decoding. First-token probability prediction requires a **single** forward pass each time, while answer decoding involves **multiple** forward passes until the model outputs an end-of-sequence token. Since ICN modifies the probing process by adding (#Query-1) additional first-token probability predictions, its time overhead remains relatively small.
> > >
> > > Example Calculation: For a task probing 100 answers, with an average of 3 forward passes required for answer decoding, adding ICN with #Query=10 results in a time cost of:
> > >
> > > $\text{Time Cost} = \frac{100 \cdot 3 + (10 - 1) \cdot 1}{100 \cdot 3} = 1.03$
> > >
> > > This represents only a 3% increase, which is acceptable given the precision improvements demonstrated in our experiments.
> > >
> > > Experiment: In practice, the theoretical efficiency may be influenced by the increased context length resulting from appending explored answers. To validate this, we reran the ICN experiments for knowledge probing and calculated the average time cost across multiple trials. These experiments were conducted on a single NVIDIA A100 GPU.
> > >
> > > |# Query|Extra Time (s)|Extra Time (%)|
> > > |-------------|--------|--------|
> > > |1| 0 | 0 |
> > > |2| 0.05 | 0.57% |
> > > |5| 0.15 | 1.71% |
> > > |10| 0.33 | 3.78% |
> > > |25| 0.86 | 9.85% |
> > > |50| 1.87 | 21.42% |
> > >
> > > *(The average answer decoding time cost for 50 answers is 8.73s)*
> > >
> > > We observe that even when the number of queries (#Query) matches the answer count (i.e., 50), the additional time required for probing is only 21.42%. In practice, for the best-performing configurations (#Query = 5 or 10), the extra time cost ranges from just 1.71% to 3.78%, which is relatively minor. It is worth noting that knowledge probing typically involves generating very short answers. For tasks requiring longer outputs, such as text generation or chain-of-thought sampling, the additional time cost of applying ICN becomes an even smaller fraction of the overall time.
> > >
> > > **Prior Work**
> > >
> > > We deeply acknowledge the previous efforts made in addressing challenges associated with fixed unembedding parameters (vocabulary embeddings). Building on this foundation, our work introduces several substantial contributions:
> > >
> > > - Clustering Analysis: We employ a clustering algorithm to reveal how vocabulary embeddings correlate, uncovering patterns such as groups of characters, numbers, and prefixes.
> > > - Misalignment with Decoding Objectives: We demonstrate how these correlations conflict with the goal of achieving diverse language model decoding.
> > > - Fine-Tuning-Free Solution: To address this misalignment, we propose a novel method that leverages contextual information to eliminate probabilities of probed answers without requiring fine-tuning.
> > > - Beyond inference, our work also sheds light on the fine-tuning process. Specifically, we expose limitations in generalization by predicting the next tokens, offering new insights into learning language models.
> > >
> > > These contributions set our work apart, underscoring its originality and impact in advancing the field.

---

> ### Author Response · Authors · 2024-11-24
>
> Dear reviewer ciXY, we sincerely appreciate your valuable feedback and have made the necessary clarifications and updates based on your suggestions. Thank you for your efforts in helping us further refine our work, especially in its connection to previous works!
>
> As the discussion period approaches its conclusion, we kindly ask if you could spare some time to review our rebuttal and evaluate whether it addresses your concerns. We would be delighted if you find our responses satisfactory and consider adjusting your rating accordingly. If there are any remaining questions or concerns, we remain actively available to address them promptly.
>
> Thank you once again for your time and thoughtful input!

---

> > ### Author Response · Authors · 2024-11-26
> >
> > Good news! The discussion period has been extended by 6 days. Please don't hesitate to express your potential confusion about the details of our work and response. We are looking forward to more valuable feedback from you. We will stay passionate to address them for you, which will also benefit our work. Thanks for the time and effort you spend on our work!

---

> ### Author Response · Authors · 2024-12-02
>
> Dear Reviewer ciXY,
>
> We enjoyed the discussion with you during the rebuttal period, which is valuable to the literature contextualization of our paper. As the discussion period finally approaches its end, we hope our latest response has fully addressed your remaining concerns such as efficiency. If you have any further questions, please feel free to share them with us! In the last two days, we will continue showing our greatest respect for your devotion to the discussion by staying active for any questions. Thanks again for your time and thoughts!

---

### Meta-Review · Area_Chair_Ni19 · 2024-12-20

**Metareview:**

The paper addresses the problem that traditional decoding methods do not adequately explore different regions of the vocab space. To address this, the authors propose an in-context learning (prompting) method to push away from the explored regions.

Reviewers generally believe the paper is interesting and the results are good. However, concerns are raised regarding the lack of discussion on previous work, such as softmax bottleneck. The authors have updated their paper addressing the concerns.

**Additional Comments On Reviewer Discussion:**

Reviewers generally believe the paper is interesting and the results are good. However, concerns are raised regarding the lack of discussion on previous work, such as softmax bottleneck. The authors have updated their paper addressing the concerns.

---

### Decision · Program_Chairs · 2025-01-22

Accept (Poster)